# The Sparse Matrix-Based Random Projection: A Study of Binary and Ternary Quantization

**Weizhi Lu**                                           *wzlu@sdu.edu.cn*
*School of Control Science and Engineering, Shandong University*
*Key Laboratory of Machine Intelligence and System Control, Ministry of Education*

**Zhongzheng Li**                                 *lizhongzheng@mail.sdu.edu.cn*
*School of Control Science and Engineering, Shandong University*

**Mingrui Chen**                                      *mrchen@mail.sdu.edu.cn*
*School of Control Science and Engineering, Shandong University*

**Weiyu Li**[*]                                          *liweiyu@sdu.edu.cn*
*Zhongtai Securities Institute for Financial Studies, Shandong University*
*National Center for Applied Mathematics in Shandong*

**Reviewed on OpenReview:** *https://openreview.net/pdf?id=dNJmJ8bh1M*

## Abstract

Random projection is a simple yet effective technique for dimension reduction, widely used in various machine learning tasks. Following the projection step, quantization is often applied to further reduce the complexity of projected data. In general, quantized projections are expected to approximately preserve the pairwise distances between the original data points, to avoid significant performance degradation in subsequent tasks. While this distance preservation property has been investigated for Gaussian matrices, our work further extends the analysis to hardware-friendly $\{0,1\}$-binary matrices, particularly focusing on cases where the projections are quantized into two types of low bit-width codes: $\{0,1\}$-binary codes and $\{0,\pm1\}$-ternary codes. It is found that the distance preservation property tends to be better maintained, when the binary projection matrices exhibit sparse structures. This is validated through classification and clustering experiments, where extremely sparse binary matrices, with only one nonzero entry per column, achieve superior or comparable performance to other denser binary matrices and Gaussian matrices. This presents an opportunity to significantly reduce the computational and storage complexity of the quantized random projection model, without compromising, and potentially even improving its performance.

## 1 Introduction

Random projection is an unsupervised dimension reduction technique (Johnson & Lindenstrauss, 1984) that involves projecting a high-dimensional data vector $x \in \mathbb{R}^n$ to a lower dimension via a linear projection

$$x' = Rx, \tag{1}$$

where $R \in \mathbb{R}^{m \times n}$ is a random matrix, $m < n$. For random matrices following Gaussian distributions (Dasgupta & Gupta, 1999), sparse $\{0,\pm1\}$-distributions (Achlioptas, 2003) and $\{0,1\}$-distributions (Dasgupta et al., 2017; Li & Zhang, 2022), it has been proved that the distance between any two original data points $x$ can be approximately preserved with high probability by their projections $x'$. This property of maintaining pairwise distances ensures the approximate preservation of the original data structure, making random projection widely applicable in various machine learning tasks without compromising significant performance.

---

[*]Corresponding author.

To further reduce the complexity of random projections, it is common practice to apply an element-wise quantization operation $f(x')$, such as the widely-used $\{0, 1\}$-binary or $\{0, \pm 1\}$-ternary quantization, to the projection $x'$ of original data $x$. This results in a *quantized* random projection model, which finds applications in various domains like the large-scale retrieval (Charikar, 2002) and deep network quantization (Wan et al., 2018; Qin et al., 2020). For the effective application of quantized random projection models, the major concern remains on the preservation of pairwise distances. Specifically, provided two data points $u$, $v \in \mathbb{R}^n$ along with their projections $u'$, $v' \in \mathbb{R}^m$, one needs to identify a random matrix $R \in \mathbb{R}^{m \times n}$ that holds the relation of $\|f(u') - f(v')\| = \|u - v\|$, or equivalently $f(u')^\top f(v') = u^\top v$ for normalized data, with high probability. While this distance preservation property $f(u')^\top f(v') = u^\top v$ has been studied for Gaussian matrices (Charikar, 2002; Li et al., 2014), it has not been explored for sparse $\{0, \pm 1\}$-ternary or $\{0, 1\}$-binary matrices. In practice, however, sparse matrices are favored due to their simpler structures. To simplify the structure of sparse matrices, there is considerable interest in estimating their sparsest form, specifically the minimum number of nonzero entries needed, while maintaining the distance preservation property described earlier. This presents a discrete optimization problem, which seems challenging to address using the probability analysis method commonly employed for Gaussian matrices. In the paper, we demonstrate that this problem can be effectively tackled, if the data for projection exhibit sparse distributions.

Sparse distributions in data are commonly encountered in signal processing and machine learning. For instance, it is well-known that natural data, such as images and sounds, usually contain coherent structures and redundant information over spatial or time domains (Ruderman, 1994; Simoncelli, 1999; Weiss & Freeman, 2007; Kotz et al., 2012; Iyer & Burge, 2019), and thus allow to be sparsified via globally or locally linear transforms, such as the discrete cosine transform (DCT) (Rao & Yip, 2014; Eude et al., 1994), the discrete wavelet transform (DWT) (Mallat, 2009), the deep convolutional neural networks (CNN) (Krizhevsky et al., 2012), and so on. In general, these sparse transforms will provide more discriminative features for classification, especially when zeroing out the small-magnitude feature elements caused by high-frequency noise (Zarka et al., 2020). Furthermore, the feature discrimination could be improved further, as the remaining large feature elements are quantized to the values of $\pm 1$ or 1 through appropriate ternary or binary quantization (Lu et al., 2023). This suggests that employing low bit-width binary and ternary quantization on sparse features is advantageous for classification in terms of both complexity and accuracy. Then for the quantized random projection of sparse features, instead of the conventional distance preservation property of $f(u')^\top f(v') = u^\top v$, we propose to investigate the property of $f(u')^\top f(v') = f^\top(u)f(v)$, i.e. preserving the distance between the quantization codes $f(u)$ of sparse features $u$, in order to allow the quantized projections $f(u')$ to obtain more discriminative features from the quantized original data $f(u)$.

With the quantized sparse features as input, the random projection model is somewhat similar to the compressed sensing model (Donoho, 2006). Inspired by the analysis of the sparse $\{0, 1\}$-binary matrix-based compressed sensing (Mendoza-Smith & Tanner, 2017; Lu et al., 2018), in the paper we investigate the proposed distance preservation property $f(u')^\top f(v') = f^\top(u)f(v)$ for the sparse binary matrix-based random projection. By varying the matrix sparsity, we observe that the property tends to be better satisfied by extremely sparse matrices containing only one nonzero entry per column, compared to other denser matrices. As expected, theses extremely sparse matrices also perform better in both supervised and unsupervised learning tasks, specifically classification and clustering. This property is highly attractive for random projection in terms of both complexity and accuracy. Overall, the major contributions of the paper can be summarized as follows.

- For the binary matrix-based random projection, we for the first time study the impact of matrix sparsity on the performance of ternary (and binary) *quantized* projections in the conventional classification and clustering tasks. It is found that extremely sparse binary matrices that contain only one nonzero entry per column tend to outperform other denser matrices, when the original data for projection consist of commonly-used sparse features, such as DWT and CNN features derived from the well-known datasets like YaleB (Georghiades et al., 2001; Lee et al., 2005), CIFAR10 (Krizhevsky & Hinton, 2009) and ImageNet (Deng et al., 2009).

- To estimate the optimal matrix sparsity, we investigate how accurately the ternary (and binary) quantized projection can preserve the pairwise distance between the ternary (and binary) quanti-

zation of original data, rather than directly between the original data as conventionally studied. Compared to the conventional distance preservation property, our proposed property offers two key advantages. First, it allows the quantized projection to obtain more discriminative features from the original data, which are the sparse features as previously mentioned. Second, it facilitates easier analysis for the quantized, binary matrix-based random projection model, whereas the conventional distance preservation property poses greater challenges.

The rest of the paper is organized as follows. In the next section, we review the literature related to the quantized random projection model. In Section 3, we introduce the basic knowledge about the model and describe the proposed distance preservation property. Among the binary matrices with different sparsity, the one that better holds the proposed property is estimated in Section 4. The performance advantage of such matrix in classification and clustering is verified in Section 5. Section 6 concludes the work.

## 2    Related work

The quantized random projection model has been studied in two research areas: local similarity hashing (LSH) (Charikar, 2002; Boufounos & Rane, 2013; Valsesia & Magli, 2016) and compressed sensing (Jacques et al., 2013). The former aims to adopt quantized projections to build hash tables for information retrieval, and the latter aims to reconstruct original data from quantized projections. Different from our work, both of them, broadly speaking, require the quantized projection $f(x')$ to preserve the pairwise distance (or similarity) between original data $x$, rather than between their quantization versions $f(x)$. Furthermore, their studies are mainly focused on Gaussian matrices. For the classification on quantized projections, a systematic evaluation has been presented in (Li et al., 2014), which demonstrates that compared to unquantized projections, a slight performance reduction inclines to be caused by 2-bit quantization, and the reduction becomes noticeable for 1-bit quantization. Recent empirical studies have demonstrated that the binary and ternary quantization can improve classification accuracy, when applied to commonly-used sparse features (Lu et al., 2023). Moreover, when classifying quantized projections of sparse features, random projections based on extremely sparse $\{0, \pm 1\}$-matrices generally outperform those based on Gaussian matrices. This remarkable performance motivates us to conduct a thorough theoretical analysis to explore the impact of sparse matrices on the classification of quantized projections.

For random projections based on sparse matrices, like $\{0, \pm 1\}$-ternary matrices and $\{0, 1\}$-binary matrices, existing research mainly explores the distance preservation property for the linear model (1), without quantization considered. Specifically, the $\ell_2$ distance preservation property of ternary matrices has been studied in (Li et al., 2006), which demonstrates that the property can be well satisfied when the matrix has the proportion of nonzero entries greater than $1/\sqrt{n}$. In (Dasgupta et al., 2017), the $\ell_2$ distance preservation is analyzed for binary matrices, and empirically the matrices tend to reach a stable performance for nearest neighbors search when containing more than about 10% nonzero entries. In contrast, our study demonstrates that for the quantized projections of sparse features, binary matrices can generally achieve the best classification performance when containing only one nonzero entry per column.

## 3    Problem Formulation

In the paper, we study the random projection model (1) which has the original data $x \in \mathbb{R}^n$ sparsely distributed and has the random matrix $R \in \{0, 1\}^{m \times n}$ binary distributed. To improve the classification on the quantization of projected data, we present a novel distance preservation property that maintains the pairwise distance between the quantization of original data, rather than between the original data themselves, and then investigate the probability that the property holds for the binary matrix with varying matrix sparsity. In this section we provide the basic knowledge about the study, including the distribution of binary matrices $R$, the distribution of original data $x$, the quantization functions $f(\cdot)$, as well as the distance preservation property.

### 3.1 Binary matrices

For a random binary matrix $R \in \{0,1\}^{m \times n}$, we assume it contains $d$ $(< m)$ nonzero entries per column, or say having column degree $d$. This parameter measures the matrix sparsity, whose impact on distance preservation will be the core of our research. We denote $R_{i,j} \in \mathbb{R}$ as the entry at the $i$-th row and $j$-th column, $R_{*,j} \in \mathbb{R}^m$ the $j$-th column vector, $R_{i,*} \in \mathbb{R}^{1 \times n}$ the $i$-th row vector, $R_{i,\phi} \in \mathbb{R}^{1 \times |\phi|}$ the intersection of the $i$-th row and the columns indexed by $\phi \subset [n]$, $[n] := \{1, 2, ..., n\}$, and $R_{*,\phi} \in \mathbb{R}^{m \times |\phi|}$ the set of the columns indexed by $\phi$. Moreover, inspired by the analysis of the binary matrix-based compressed sensing (Donoho, 2006), in Definition 1 we model the adjacency relation between the binary matrix's rows and columns, which corresponds to the mapping relation between the coordinates of original data $x$ and projected data $x'$. The relation will be explored in the following distance preservation analysis.

**Definition 1** (Adjacency relation between the binary matrix's rows and columns)**.** Consider the binary matrix $R \in \{0,1\}^{m \times n}$ with its columns and rows indexed by the variables $j$ and $i$, respectively. For the matrix's $j$-th column, define its adjacent row set as $\mathcal{N}(j) = \{i : R_{i,j} \neq 0, i \in [m]\}$; and subsequently, for a subset of the columns $J \subset [n]$, define its adjacent row set as $\mathcal{N}(J) = \{\bigcup_j N(j), j \in [J]\}$. Similarly, for the matrix's $i$-th row, define its adjacent column set as $\mathcal{N}(i) = \{j : R_{i,j} \neq 0, j \in [n]\}$. Notice that the matrix's columns and rows correspond respectively to the element coordinates of the original data $x$ and projected data $x'$, and so the adjacency relation defined above can be used to describe the mapping relation between the coordinates of the two kinds of data.

### 3.2 Original data

The analysis of the quantized random projection is related to the distribution of the original data $x = (x_1, x_2, \cdots, x_n)^\top \in \mathbb{R}^n$. In the paper, we propose to study the data with approximately sparse or exactly sparse distributions, as specified in Definitions 2 and 3.

**Definition 2** (Approximately sparse data)**.** A data vector $x \in \mathbb{R}^n$ is called approximately sparse, if its element-magnitude-ordered version $x^* = (x_1^*, x_2^*, \cdots, x_n^*)$ follows an exponential decay relation: $|x_{i+1}^*|/|x_i^*| \leq e^{-\beta}$, where $\beta$ is an arbitrary positive constant; and the larger the value of $\beta$, the faster the decaying speed.

**Definition 3** (Exactly sparse data)**.** A data vector $x \in \mathbb{R}^n$ is called $k$ sparse, or having sparsity $k$, if it contains exactly $k$ $(\ll n)$ nonzero entries, or say having the support size $|supp(x)| = k$, $supp(x) = \{i : x_i \neq 0, i \in [n]\}$.

The approximately sparse data are prevalent in various classification tasks, such as the features extracted with DCT, DWT, CNN and so on. It is known that these features have approximately sparse distributions, and can be modeled with exponential decay functions (Weiss & Freeman, 2007; Kotz et al., 2012). Moreover, they can be further transformed to exactly sparse structures by zeroing out the elements of small magnitude. Compared to approximately sparse structures, exactly sparse structures have three advantages. First, it can reduce the computation complexity of the downstream random projection operation. Second, as studied in (Lu et al., 2023), it tends to improve feature discrimination, favorable for classification. Third, as detailed later, it is easier to analyze due to its simpler distributions, and it allows us to simply set the projection's quantization threshold to a constant value, zero, to achieve the desired performance. Therefore, in the final experiments we will pay more attention to the performance of exactly sparse features.

### 3.3 Quantization functions

We investigate two fundamental, element-wise quantization operations: the binary and ternary quantization. The ternary quantization is formulated as

$$f_\tau(x_i) = \begin{cases} +1, & x_i > \tau \\ -1, & x_i < -\tau \\ 0, & \text{others} \end{cases} \tag{2}$$

where the threshold parameter $\tau \geq 0$ is empirically determined to regulate the sparsity of the quantization $f_\tau(x)$ for the vector $x \in \mathbb{R}^n$. We treat $f_\tau(\cdot)$ as an element-wise function and express the vector's quantization

as $f_\tau(x) = (f_\tau(x_1), f_\tau(x_2), \cdots, f_\tau(x_n))^\top$. Similarly, the $\{0, 1\}$-binary quantization can be formulated using a single threshold parameter $\tau$. For brevity, the following analysis will focus on ternary quantization, but the results can be easily extended to binary quantization.

### 3.4 Distance preservation property

Consider the random projection model (1), which has two original data $u, v \in \mathbb{R}^n$ and corresponding projections $u', v' \in \mathbb{R}^m$. We aim to determine the distribution of the binary matrix $R$ that holds the following distance preservation property

$$f_{\tau_3}(u')^\top f_{\tau_4}(v') = \alpha \cdot f_{\tau_1}^\top(u) f_{\tau_2}(v) \tag{3}$$

with high probability, where $\alpha$ is a positive constant, and the threshold parameters $\tau_i$ of the quantization functions $f(\cdot)$ will be determined by analysis. Notice that for the convenience of analysis, the parameter $\alpha$ is introduced to define a relative distance preservation, whose value varying does not affect the classification of projected data; and the exact distance preservation, namely the case of $\alpha = 1$, can be easily obtained by scaling the element values of random matrices.

Different from the traditional quantized random projection model that requires preserving the distance between two original data $u$ and $v$, our proposed distance preservation model (3) maintains the distance between the two original data's quantization codes, $f_{\tau_1}(u)$ and $f_{\tau_2}(v)$. This proposal is inspired by the recent finding (Lu et al., 2023) that the quantization of sparse features (i.e. our original data) can produce more discriminative features for classification. Then compared to the conventional distance preservation, the proposed distance preservation (3) allows the projection to capture more discriminative features of the original data; and moreover, the proposed method is easier to analyze, since its quantization operations simplify the modeling of data distributions.

## 4 Distance preservation analysis

For the projection matrix $R \in \{0, 1\}^{m \times n}$ with varying column degree $d$, in this section we estimate the optimal column degree $d$ that ensures the proposed distance preservation property (3) holding with high probability. For ease of analysis, we first describe the desired matrix structure that holds the property (3) for two given data $x \in \mathbb{R}^n$, and then derive the probability that the desired matrix structure holds for two arbitrary data $x \in \mathbb{R}^n$. The analysis results are presented in Theorems 1-3, with comprehensive proofs outlined in Appendices A.1-A.3. For brevity, we mainly analyze the ternary quantized projections $f_\tau(x')$, and the analysis can be straightforwardly extended to the binary case.

### 4.1 Distance preservation for two given data

Given two original data points $u, v \in \mathbb{R}^n$ with deterministic structures, we evaluate the distance preservation property in Theorems 1 and 2, respectively for two different data distributions: exactly sparse and approximately sparse, as specified in Definitions 3 and 2. On the whole, both theorems demonstrate that the proposed distance preservation property (3) will be achieved, if the submatrix $R_{*,\phi}$ of the binary matrix $R$, indexed by the support union $\phi$ of the two quantization codes $f_{\tau_1}(u)$ and $f_{\tau_2}(v)$ corresponding to the two original data points, has orthogonal columns. The details are discussed in their respective remarks.

**Theorem 1** (Exactly sparse data). Consider the random projection model (1), which has two projected data $u', v' \in \mathbb{R}^m$ generated from two exactly sparse data $u, v \in \mathbb{R}^n$ with sparsity $k_1, k_2$, provided a random matrix $R \in \{0, 1\}^{m \times n}$ with column degree $d$ $(< m)$. Let $\phi = supp(u) \cup supp(v)$, then $|\phi| \le k_1 + k_2$. If $R_{*,\phi}^\top R_{*,\phi} = dI_{|\phi|}$, where $I_{|\phi|}$ denotes the identity matrix of size $|\phi|$, we have

$$f_0(u')^\top f_0(v') = d \cdot f_0(u)^\top f_0(v), \tag{4}$$

where $f_0(\cdot)$ is the ternary quantization function (2) with parameter $\tau = 0$.

**Remark of Theorem 1.** For the theorem, there are several noteworthy points. (i) The orthogonal $R_{*,\phi}$ required by the theorem will be obtained, as the support union size of two original data is less than the

matrix's row size, that is $|\phi| \leq m$. With a given column degree $d$, statistically, the orthogonal $R_{*,\phi}$ is more likely to be obtained when its row size $m$ is large, and the column size is small, corresponding to lower data sparsity $k_i$. (ii) Exactly sparse data with small sparsity $k_i$ can be obtained by zeroing out the small-magnitude elements of sparse features. As previously mentioned, this sparsifying operation can improve feature discrimination, beneficial for classification (Lu et al., 2023). (iii) The four ternary functions in (4) all simply fix their threshold parameter to $\tau = 0$ for both the original data and projected data, eliminating the need of parameter tuning. (iv) Since the sparsity of exactly sparse data remains unchanged after ternary or binary quantization, we can directly use their quantization codes for projection, without affecting the final projection results (4). This suggests that Theorem 1 holds for the random projection model where both the original data and projected data are quantized into ternary or binary codes.

**Theorem 2** (Approximately sparse data). Consider the random projection model (1), which has two projected data $u'$, $v' \in \mathbb{R}^m$ generated from two approximately sparse data $u$, $v \in \mathbb{R}^n$, provided a random matrix $R \in \{0,1\}^{m \times n}$ with column degree $d$. For $u$ and $v$, assigning two ternary functions $f_\tau(\cdot)$ with $\tau = \tau_1 = \frac{|u^*_{k_1}| + |u^*_{k_1+1}|}{2}$ and $\tau = \tau_2 = \frac{|v^*_{k_2}| + |v^*_{k_2+1}|}{2}$, respectively, such that $supp(f_{\tau_1}(u)) = k_1$ and $supp(f_{\tau_2}(v)) = k_2$, where $u^*_{k_1}$ denotes the $k_1$-th largest element of $u$ in magnitude and $v^*_{k_2}$ is defined similarly. Let $\phi = supp(f_{\tau_1}(u)) \cup supp(f_{\tau_2}(v))$, then $|\phi| \leq k_1 + k_2$. If $R^\top_{*,\phi} R_{*,\phi} = dI_{|\phi|}$ and $u$, $v$ have their decaying parameter $\beta \geq \ln(2 + \sqrt{3})$, we can derive that

$$f_{\tau_1}(u')^\top f_{\tau_2}(v') = d \cdot f_{\tau_1}(u)^\top f_{\tau_2}(v). \tag{5}$$

**Remark of Theorem 2.** (i) The results we have derived for approximately sparse data are similar to those obtained for exactly sparse data by Theorem 1. One of major differences between them is the choice of the threshold parameter $\tau$ for ternary functions. As discussed in Section 3.4, we need to select a proper $\tau$ to produce a data sparsity $k$ that can improve feature discrimination when transforming $u$ to $f_\tau(u)$, thus leading to better classification performance. As shown in (Lu et al., 2023), the desired sparsity $k$ can be empirically determined. Without loss of generality, we assume two different sparsity values $k_1$, $k_2$ (corresponding to $\tau_1$ and $\tau_2$) for the two original data points $u$, $v$, in order to obtain the desired quantization performance. In practical applications, for simplicity, we suggest to select a same sparsity $k$ for the two data, since they are generally obtained from the same scene and share similar distributions. (ii) Moreover, it is worth noting that besides the orthogonal constraint on the submatrix $R_{*,\phi}$, the derivation of (5) also imposes a constraint on the distribution of the original sparse data: the data should have its decaying parameter $\beta \geq \ln(2 + \sqrt{3})$, and roughly speaking, the data needs to decay sufficiently fast. Notice that the lower bound for $\beta$ is a sufficient but not necessary condition, and empirically our optimal matrix estimation is not sensitive to the lower bound of $\beta$ and tends to achieve the desired classification performance even for the sparse features with smaller $\beta$.

## 4.2 Distance preservation for two arbitrary data

To generalize the distance preservation property (3) from two fixed data to arbitrary data, we should extend the condition of orthogonal $R_{*,\phi}$ from a fixed column set $\phi = supp(f_{\tau_1}(u)) \cup supp(f_{\tau_2}(v))$ to an arbitrary set $\phi \subset [n]$, $|\phi| = k_1 + k_2 < m$. For a randomly generated binary matrix $R \in \{0,1\}^{m \times n}$, however, it is hard to ensure its each submatrix $R_{*,\phi}$ to have orthogonal columns. In Theorem 3, we analyze the probability of having orthogonal $R_{*,\phi}$ under the varying column degree $d$.

**Theorem 3.** Given a random matrix $R \in \{0,1\}^{m \times n}$ with column degree $d$. Consider its submatrix $R_{*,\phi}$ with $\phi \subset [n]$. Denote $Pr\{R^\top_{*,\phi} R_{*,\phi} = dI_{|\phi|}\}$ as the probability that $R^\top_{*,\phi} R_{*,\phi} = dI_{|\phi|}$ holds for any $\phi \subset [n]$, with $|\phi| \geq 2$ and $d|\phi| \leq m$. Provided $m$ and $\phi$, we have the probability

$$Pr\{R^\top_{*,\phi} R_{*,\phi} = dI_{|\phi|}\} = \frac{[(m-d)!]^{|\phi|}}{(m!)^{(|\phi|-1)}(m - |\phi|d)!} \tag{6}$$

$$\leq \frac{\prod_{\ell=0}^{|\phi|-1}(m-\ell)}{m^{|\phi|}} \tag{7}$$

which has the value of (6) monotonically decreasing with the column degree $d$, and has the equality of (7) attained at $d = 1$.

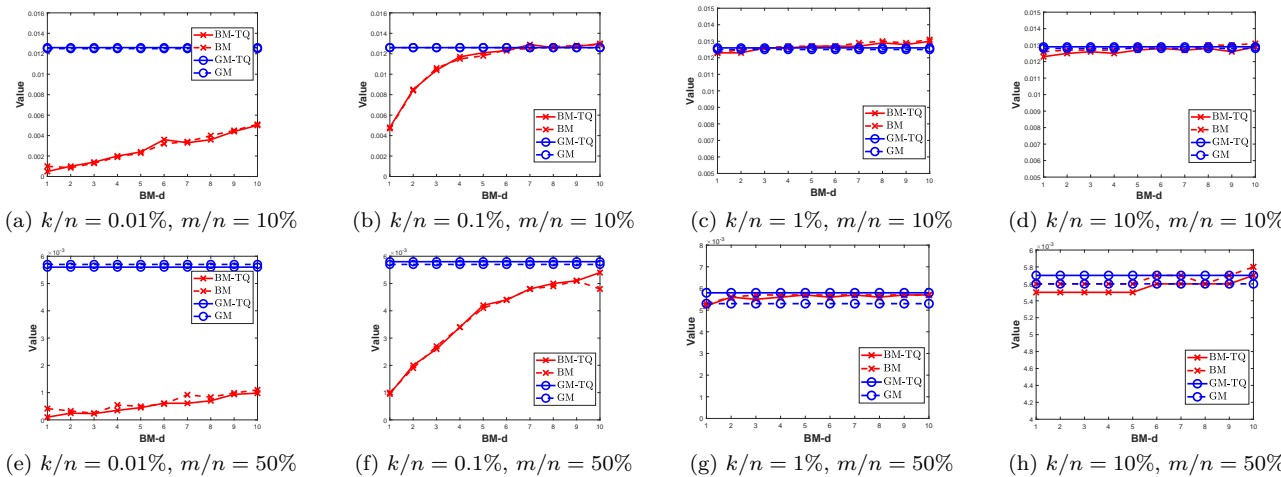

Figure 1: Distance variation rate for the ternary-quantized (TQ) (and non-quantized) projections of the generated data, with four different feature sparsity ratios $k/n = 0.01\%$, $0.1\%$, $1\%$ and $10\%$, using two projection matrices: the Gaussian matrix (GM) and the binary matrix (BM) with varying column degree BM-d $\in [1, 10]$, under two projection ratios $m/n = 10\%$ and 50%. Note the smaller the distance variation rate, the better the distance preservation.

**Remark of Theorem 3.** (i) The theorem demonstrates that the probability (6) of having orthogonal $R_{*,\phi}$ will increase with the decreasing of column degree $d$. It suggests that the distance preservation property (3) should be satisfied with higher probability by the binary matrix with smaller column degree $d$. Then it is reasonable to conjecture that when applied to the common classification or clustering tasks, quantized projections can achieve the best performance with very sparse binary matrices, i.e. the ones with column degree as small as $d = 1$, as verified in our experiments. (ii) Moreover, it is worth noting that besides the column degree $d$, the probability (6) is also related to the size of $\phi$. For the probability derived with $d = 1$ in (7), it is easy to see that the smaller the $|\phi|$ value, the higher the probability. This means that the sparser features $x$ (with smaller sparsity $k$) should result in the better distance preservation property (3). This relation is similar to the condition of compressed sensing (Donoho, 2006).

### 4.3 Extension to binary quantization

In Theorems 1 and 2, we only investigate the ternary quantization (2) for the distance preservation property (3). From the proofs of the two theorems, it can be seen that their results can be directly extended to the case of binary quantization, with the same threshold values $\tau_i$. Then by Theorem 3, we can predict that the binary quantization of projected data will achieve its best classification performance when using very sparse binary matrices. This is verified in our experiments. In the paper, we pay more attention to ternary quantization than to binary quantization, as the latter generally performs worse due to discarding more feature elements (Lu et al., 2023).

### 4.4 Numerical validation

In this part, we aim to validate the main result of Theorem 3, that is the proposed distance preservation property (3) should be satisfied with higher probability by binary matrices with smaller column degrees $d$, when the sparsity $k$ of the quantization $f_{\tau_i}(x)$ of original data $x$ is sufficiently small, as required in Theorems 1 and 2. For this purpose, we directly investigate the distance variation before and after random projections through numerical simulation. Based on the distance preservation property (3), we introduce the concept of the *distance variation rate*, which quantifies the relative change in pairwise distance between quantized *projected* data compared to the distance between quantized *original* data. Specifically, it is defined as

$$\frac{1}{N} \sum_{i=1}^{N} \frac{\left| \|f_{\tau_3}(u^{(i)\prime}) - f_{\tau_4}(v^{(i)\prime})\|_2 - \|f_{\tau_1}(u^{(i)}) - f_{\tau_2}(v^{(i)})\|_2 \right|}{\|f_{\tau_1}(u^{(i)}) - f_{\tau_2}(v^{(i)})\|_2} \tag{8}$$

where $u^{(i)}$, $v^{(i)} \in \mathbb{R}^n$ are a pair of original data points, $1 \leq i \leq N$, and $u^{(i)\prime}$, $v^{(i)\prime} \in \mathbb{R}^m$ are their projections over a random matrix $R \in \{0, 1\}^{m \times n}$ with column degree $d$. Note that all the quantized data $f_{\tau_i}(\cdot)$ in (8) are pre-normalized using $\ell_2$ norm, in order to eliminate the magnitude discrepancy between before and after random projections. It is evident that the smaller the distance variation rate (8), the better the distance preservation property (3).

By Theorems 1 and 2, given a column degree $d$, a better distance preservation property (3) tends to be achieved with a smaller sparsity ratio $k/n$ in the quantization $f_{\tau_i}(u^{(i)})$ of original data $u^{(i)}$, and a larger projection ratio $m/n$ for random matrices. To generate the original data $u^{(i)}$ flexibly with any given $k$, namely having $\left| supp(f_{\tau_i}(u^{(i)})) \right| = k$, we simply set $\tau = 0$ and ensure the number of nonzero entries in $u^{(i)}$ equal to $k$. Considering the specific values of the nonzero entries do not affect the distance preservation property, as demonstrated in Theorems 1 and 2, we set $u^{(i)} \in \{0, \pm 1\}^n$ in the case of ternary quantization $f_{\tau_i}(\cdot)$ for easy simulation. Moreover, we set the number of data pairs $N = 10000$, the original data dimension $n = 10000$, the data sparsity ratio $k/n \in \{0.01\%, 0.1\%, 1\%, 10\%\}$, the matrix's column degree $d \in [1, 10]$, and the projection ratio $m/n \in \{10\%, 50\%\}$.

The simulation results are provided in Figure 1. For comparison, the results for the popular Gaussian matrix-based random projection are also provided. Figure 1 illustrates that as expected in Theorem 3, the distance variation rate of random binary matrices inclines to increase with the column degree $d$. This indicates a decline in distance preservation capability. This trend is particularly evident, when the sparsity ratio $k/n$ of original data is relatively small, such as $k/n < 1\%$. This is consistent with our theoretical analysis. As the sparsity ratio $k/n$ increases, binary matrices tend to exhibit similar distance preservation performance across different column degrees $d$. Compared to Gaussian matrices, binary matrices can often achieve lower distance variation rates, indicating better distance preservation performance, especially for small values of $k/n$. As $k/n$ increases, two kinds of matrices tend to exhibit comparable distance preservation performance. These trends are also observed in the case of binary quantization, where both the original and projected data are quantized to $\{0, 1\}$-binary values, as detailed in Appendix A.4.1, Figure 7. Notably, the above performance trends regarding distance preservation are corroborated in the subsequent experiments conducted on classification and clustering, providing further validation of our theoretical findings.

## 5 Experiments

### 5.1 Settings

In this section, we investigate the performance of the ternary and binary-quantized projections of sparse data in both supervised and unsupervised learning tasks, specifically classification and clustering. Random projections are implemented using random binary matrices with different column degrees. Our goal is to find the column degree that leads to the best classification or clustering performance. For comparison, the performance is also examined for the popular Gaussian matrix-based random projections and for the non-quantized projections. Considering both the ternary and binary-quantized projections commonly exhibit similar performance trends with the varying of the column degree of binary matrices, we will mainly discuss the results of ternary projections and defer the results of binary projections to Appendix A.4.

#### 5.1.1 Classification and clustering algorithms

To directly reflect the impact of the distance between projected data on classification and clustering, we employ linear similarity metrics-based algorithms for both tasks. Specifically, classification is implemented with two fundamental classifiers: the K-nearest neighbor (KNN) classifier based on cosine distance (Peterson, 2009) and the support vector machines (SVM) with a linear kernel (Cortes & Vapnik, 1995). Both classifiers have performance fully dependent on the distance between data, without involving additional operations to further improve data discrimination. Empirically, the two classifiers tend to show similar performance trends as the column degree of binary matrices varies. For brevity, we will focus on the results of KNN and present the results of SVM (Cortes & Vapnik, 1995) in Appendix A.4. Clustering is implemented with the k-means algorithm based on cosine distance (MacQueen et al., 1967). To evaluate clustering accuracy, we follow the strategy adopted in (Xu et al., 2004). Given a set of labeled data, we first remove their labels to

run the clustering algorithm, then label each resulting cluster with the majority class according to the labels of original data, and calculate the proportion of the data samples correctly classified by each cluster.

### 5.1.2 Data

The sparse data intended for projection are generated from the datasets YaleB (Georghiades et al., 2001; Lee et al., 2005), CIFAR10 (Krizhevsky & Hinton, 2009) and Mini-ImageNet (Vinyals et al., 2016), respectively via the feature transforms DWT (Mallat, 2009), AlexNet Conv5 (Krizhevsky et al., 2012) and VGG16 Conv5_3 (Simonyan & Zisserman, 2014). To provide relatively good classification performance, we assign more advanced feature transforms to more complex datasets. The datasets are briefly introduced as follows. YaleB contains the face images of 38 persons, with about 64 samples per person. From the dataset, we randomly select 9/10 samples for training and the rest for testing. CIFAR10 consists of 10 classes of color images, with 6000 samples per class. Mini-ImageNet is a subset of ImageNet (Deng et al., 2009), which consists of 100 classes of color images, each class having 600 samples. For the latter two datasets, we use their default training and testing samples, with the ratio of 5/1. For the three datasets, we normalize the feature vectors with zero mean and unit variance, and reduce the vector dimensions several times to the order of thousands for easier simulation. The dimension reduction may decrease the classification or clustering accuracy but not influence our comparative study. To validate Theorems 1 and 2, we evaluate two kinds of sparse data that have approximately and exactly sparse distributions, respectively, as specified in Definitions 2 and 3. The approximately sparse ones are the original sparse features generated with DWT and CNN, and the exactly sparse ones are obtained by further sparsifying the features with given sparsity ratios of $k/n = 1\%$, 5%, 10% and 20%. Compared to the original, approximately sparse features, as mentioned earlier, the resulting exactly sparse features are usually more favorable for classification (Lu et al., 2023). For the random projection model (1), we test two different projection ratios: $m/n = 10\%$ and 50%.

### 5.2 Classification results

The classification results are provided in Figures 2-5 and Figure 6, respectively for the exactly sparse features and the approximately sparse features. In each figure, the first and second rows correspond respectively to the random projection cases of projection ratios $m/n = 10\%$ and 50%, and the four subfigures in each row correspond to the exactly sparse features with sparsity ratio $k/n = 1\%$, 5%, 10% and 20%. Considering the fact that exactly sparse features generally outperform approximately sparse features, and ternary quantization outperforms binary quantization (Lu et al., 2023), for brevity, we mainly analyze the classification on the ternary quantized projections of exactly sparse features, as illustrated in Figures 2-4, focusing on the following aspects.

### 5.2.1 Binary matrices with different column degrees

By the remark of Theorem 3, the proposed distance preservation property (3) tends to hold with higher probability, when binary matrices have a smaller column degree $d$. Then, the classification accuracy of quantized projections is expected to decrease with the increased column degree $d$. This performance trend is basically verified by the results illustrated in Figures 2-4, see the x-marked, solid lines for the classification of the ternary quantized projections of the exactly sparse features with different sparsity ratios $k/n = 1\%$, 5%, 10% and 20%. It can be seen that the performance declining speed differs with different data types, and it seems that the easier the data for classification, such as the DWT features of YaleB shown in Figure 2, the more evident the performance advantage of $d = 1$ becomes compared to other larger $d$ values. An exception worth mentioning is the case of $k/n = 1\%$, as shown in Figures 2 and 3, where $d = 1$ performs slightly worse than $d = 2$. This deviation should be attributed to the gap between theory and practice: the classification of quantized projections relates not only to the distance preservation property studied here, but also to other factors out of our scope, such as feature discrimination. Despite the imperfect, our theoretical estimation is generally supported by the results of Figures 2-4: the column degree $d = 1$ tends to provide better or comparable performance to other larger $d$, in the classification of the ternary-quantized projections of exactly sparse features.

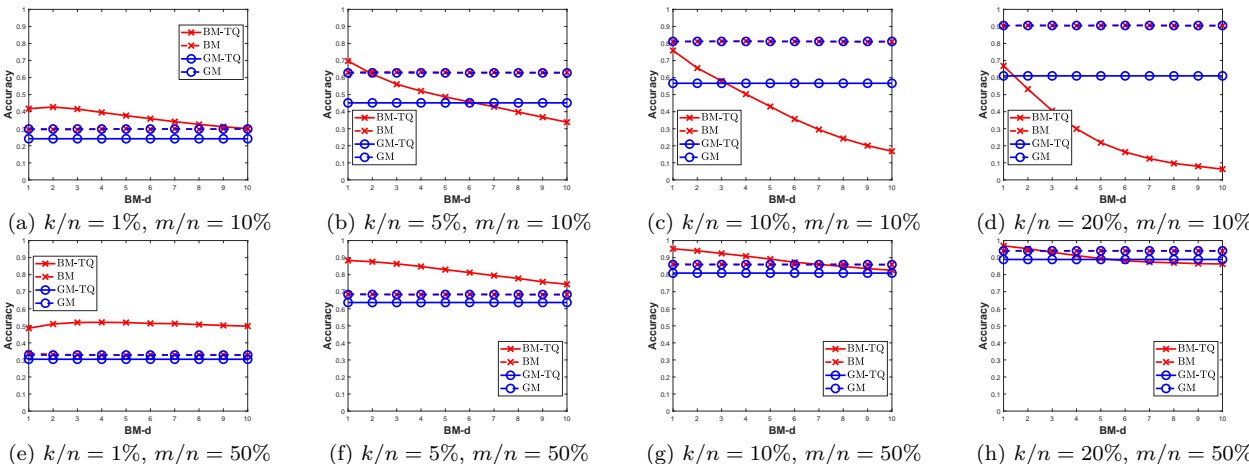

Figure 2: Classification accuracy for the ternary-quantized (TQ) (and non-quantized) projections of the exactly sparse features of YaleB (DWT), with four different feature sparsity ratios $k/n = 1\%, 5\%, 10\%$ and $20\%$, using two projection matrices: the Gaussian matrix (GM) and the binary matrix (BM) with varying column degree BM-d $\in [1, 10]$, under two projection ratios $m/n = 10\%$ and $50\%$.

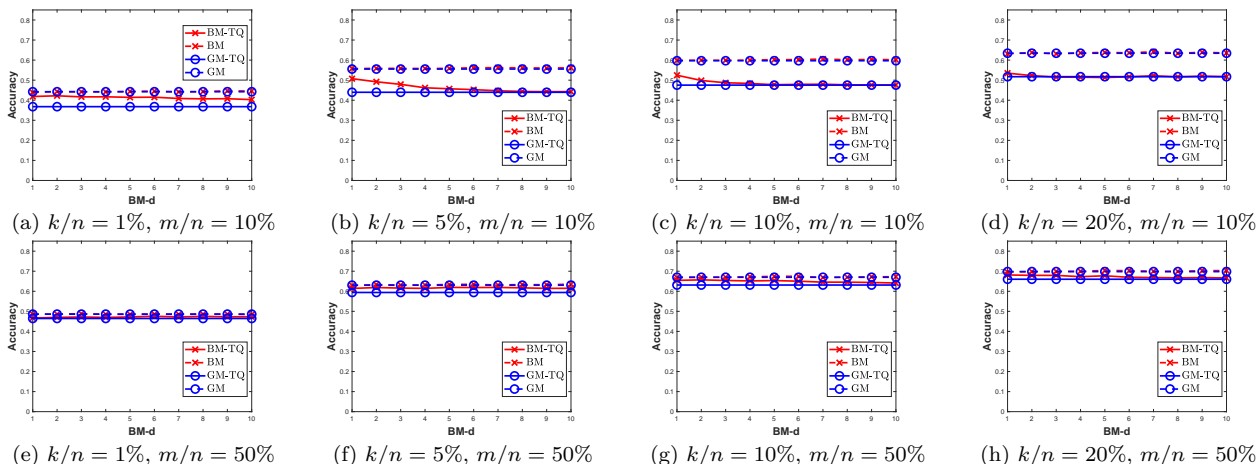

Figure 3: Classification accuracy for the ternary-quantized (TQ) (and non-quantized) projections of the exactly sparse features of CIFAR10 (AlexNet), with four different feature sparsity ratios $k/n = 1\%, 5\%, 10\%$ and $20\%$, using two projection matrices: the Gaussian matrix (GM) and the binary matrix (BM) with varying column degree BM-d $\in [1, 10]$, under two projection ratios $m/n = 10\%$ and $50\%$.

### 5.2.2 Quantized vs. non-quantized projections

By (Lu et al., 2023), quantized projections can provide better classification performance than non-quantized projections, if both the original data and random matrix have sufficiently sparse distributions, and the quantization threshold $\tau$ for projected data is properly selected. This performance property is also observed in our experiments. Comparing the classification results provided in Figures 2-4 for the ternary-quantized projections (x-marked, solid lines) and non-quantized projections (x-marked, dashed lines), it can be seen that the former tends to achieve better performance than the latter, when the column degree $d$ of binary matrix and the sparsity ratio $k/n$ of original data (i.e. the exactly sparse features) both become smaller, such as the case of $d=1$ and $k/n = 1\%$. Note that by Theorem 1 we here simply set the quantization threshold as $\tau = 0$, and a better performance for quantized projections should be obtained if the threshold is more carefully selected as in (Lu et al., 2023). Overall, the above results indicate that the sparse binary matrix with $d = 1$ can obtain better classification performance on quantized projections than on non-quantized projections. This result is highly attractive both in terms of complexity and accuracy.

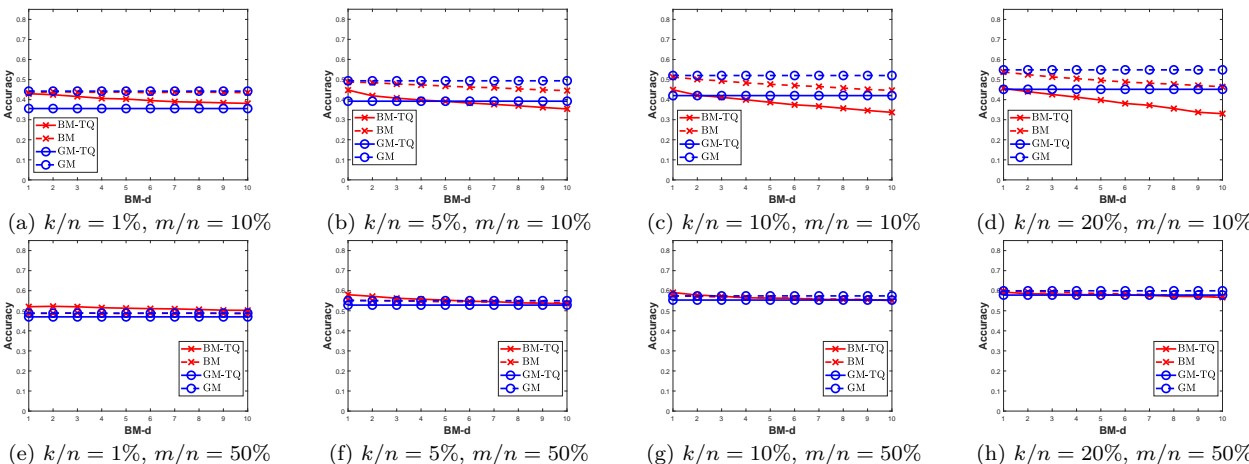

Figure 4: Classification accuracy for the ternary-quantized (TQ) (and non-quantized) projections of the exactly sparse features of Mini-ImageNet (VGG16), with four different feature sparsity ratios $k/n = 1\%$, $5\%$, $10\%$ and $20\%$, using two projection matrices: the Gaussian matrix (GM) and the binary matrix (BM) with varying column degree BM-d $\in [1, 10]$, under two projection ratios $m/n = 10\%$ and $50\%$.

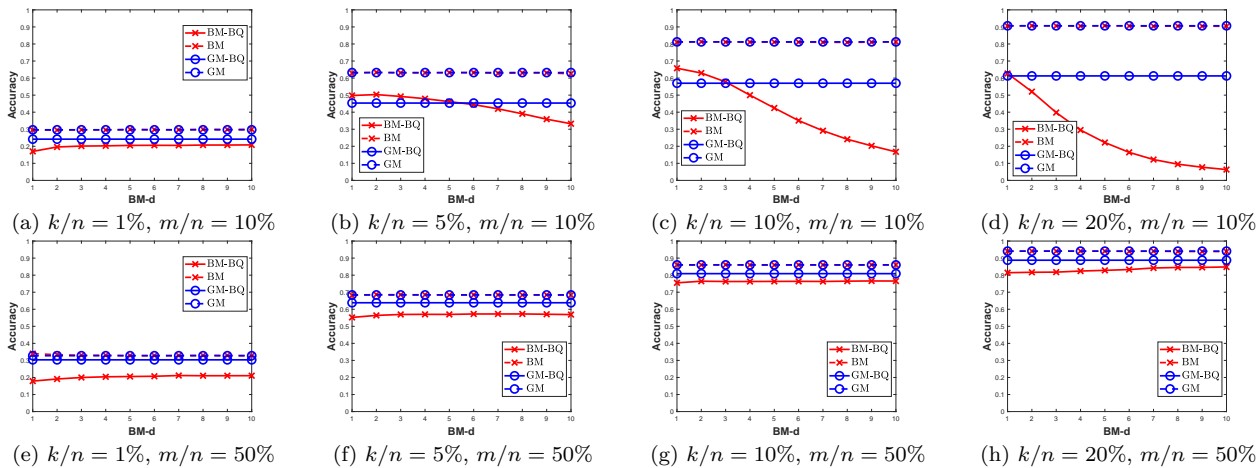

Figure 5: Classification accuracy for the binary-quantized (BQ) (and non-quantized) projections of the exactly sparse features of YaleB (DWT), with four different feature sparsity ratios $k/n = 1\%$, $5\%$, $10\%$ and $20\%$, using two projection matrices: the Gaussian matrix and the binary matrix with varying column degree BM-d $\in [1, 10]$, under two projection ratios $m/n = 10\%$ and $50\%$.

### 5.2.3 Binary matrices vs. Gaussian matrices

Figures 2-4 demonstrate that binary matrices (x-marked solid lines) tend to outperform Gaussian matrices (circle-marked solid lines), as the column degree $d$ of binary matrix and the sparsity ratio $k/n$ of original data both become smaller, such as the case of $d=1$ and $k/n = 1\%$. The superior performance of binary matrices should be attributed to its advantage in distance preservation, as demonstrated in Section 4.4. In this case, we are encouraged to replace Gaussian matrices with sparse binary matrices, for improvements both in complexity and accuracy.

### 5.2.4 Binary quantized projections

By the discussion in Section 4.3, the theoretical properties of binary matrices we derive with ternary quantized projections in Theorems 1-3 should also hold with binary quantized projections. In other words, the performance trends derived in Figures 2-4 for ternary projections, should be also achievable for binary projections. To verify this, we examine the classification on binary projections in Figure 5, see Appendix A.4.2

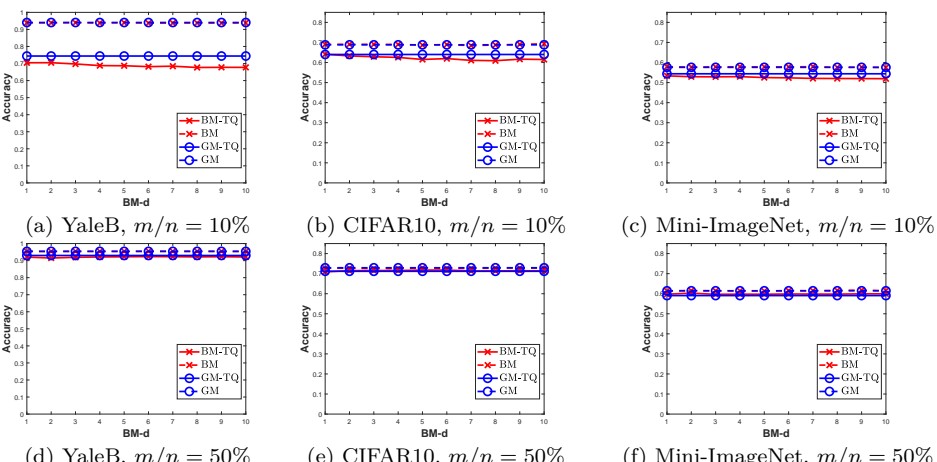

Figure 6: Classification accuracy for the ternary-quantized (TQ) (and non-quantized) projections of the original, approximately sparse features: YaleB (DWT), CIFAR10 (AlexNet), Mini-ImageNet (VGG16), using two projection matrices: the Gaussian matrix (GM) and the binary matrix (BM) with varying column degree BM-d $\in [1, 10]$, under two projection ratios $m/n = 10\%$ and 50%.

for more results. Figure 5 shows that similarly as the classification of ternary projections, in the classification of binary projections the binary matrix with column degree $d = 1$ exhibits better or comparable performance than other denser matrices. Moreover, it is worth mentioning that binary quantization often performs worse than ternary quantization, as observed in (Lu et al., 2023), due to discarding more feature elements.

### 5.2.5 Approximately sparse features

In Figure 6, we provide the classification results on the ternary quantized projections of the original features, which have approximately sparse structures. As theoretically expected, binary matrices with $d = 1$ achieve better or comparable performance to other denser matrices. Empirically, these approximately sparse features do not precisely align with the decay speed $\beta$ specified in Theorem 2. This suggests that our theoretical results demonstrate strong universal applicability, showing relatively little sensitivity to the sparsity of the original data. With the increasing of $k/n$, as illustrated in Figures 2-6, the performance advantage of binary matrices over Gaussian matrices will become less evident in the classification of quantized projections. This performance trend is consistent with the distance preservation property illustrated in Figure 1. Finally, recall that the original, approximately sparse features tend to achieve higher classification accuracy, if being further simplified to exactly sparse structures (Lu et al., 2023). Then for both better classification and easier computation, we are motivated to transform theses features to exactly sparse structures before conducting random projections on them.

### 5.3 Clustering results

Due to space limitations, we provide the clustering results for the ternary and binary quantized projections in Appendix A.4.3, specifically in Figures 11-13 and Figures 14-16, respectively. Similarly to classification, the extremely sparse binary matrix with column degree $d = 1$ exhibits several analogous properties in clustering. Specifically, 1) it can achieve superior or comparable performance to other denser binary matrices with larger $d$ values, as well as Gaussian matrices; 2) its performance on quantized projections is often better than on non-quantized projections; 3) these performance advantages can endure even when the feature sparsity ratio $k/n$ increases from 1% to 20%, gradually deviating from the theoretical condition we impose on feature sparsity.

# 6 Conclusion

For the binary matrix-based random projection, where the projected data are further quantized to binary or ternary values, we have investigated how the sparsity of binary matrices influences the ability of the quantized projections to preserve pairwise distances between the quantized original data. Our analysis indicates that binary matrices with sparser structures tend to better maintain pairwise distances, when the original data intended for projection exhibit sufficiently sparse structures. This performance trend has been validated through classification and clustering experiments on quantized projections of typical data features, including DWT features of YaleB and CNN features of CIFAR10 and ImageNet, all demonstrating approximately sparse structures. Experiments show that extremely sparse binary matrices with only one nonzero entry per column can achieve superior or comparable classification performance compared to other denser binary matrices and Gaussian matrices. The extremely sparse matrix structure allows us to significantly reduce the complexity of the quantized random projection models, such as the large-scale retrieval model (Charikar, 2002). Moreover, it is noteworthy that our research contributes to understanding and analyzing the sparse structures inherent in other advanced models that incorporate quantized random projections, like deep quantization networks (Wan et al., 2018; Qin et al., 2020), as well as biological neuron models (Dasgupta et al., 2017).

## Acknowledgements

We sincerely thank the editors and anonymous reviewers for their invaluable time and constructive feedback. Furthermore, we gratefully acknowledge the support provided by the National Key Research and Development Program of China (Grant Nos. 2022YFB3206900 and 2023YFA1008701) and the National Natural Science Foundation of China (Grant Nos. 61991412, 61801264, and 12001318).

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

# A  Appendices

## A.1  Proof of Theorem 1

*Proof.* For the two exactly sparse data points $u$, $v \in \mathbb{R}^n$, suppose their support intersection $\psi = supp(u) \cap supp(v)$. Then we can write

$$f_0(u)^\top f_0(v) = \sum_{j \in \psi} f_0(u_j) f_0(v_j). \tag{9}$$

Recall that $f_0(\cdot)$ is an element-wise function. Similarly, for the two projected points $u'$, $v' \in \mathbb{R}^m$, we define their support union and intersection as $\phi' = supp(u') \cup supp(v')$ and $\psi' = supp(u') \cap supp(v')$, and then can write

$$f_0(u')^\top f_0(v') = \sum_{i \in \psi'} f_0(u'_i) f_0(v'_i). \tag{10}$$

In the sequel, we aim to prove that (10) can be linearly transformed to (9). The analysis of (10) requires us to first determine the support intersection $\psi'$ between projected data. To achieve this, we examine the value of each element $f_0(u'_i)$ of $f_0(u')$, which for ease of analysis is divided into two groups on the basis of $i \in \mathcal{N}(supp(u))$ or not. Notice that the analysis will require us to frequently explore the adjacency relation between the random matrix's columns and rows, or say the mapping relation between the original data and projected data, as specified in Definition 1. For the case of $i \notin \mathcal{N}(supp(u))$, by Definition 1 we have $R_{i,j} = 0$, $\forall j \in supp(u)$, and then can write

$$\begin{aligned} f_0(u'_i) &= f_0 \left( \sum_{j \in [n] \setminus supp(u)} R_{i,j} u_j \right) \\ &= 0 \end{aligned} \tag{11}$$

since $u_j = 0$, $\forall j \in [n] \setminus supp(u)$; otherwise, we can derive

$$
\begin{aligned}
f_0(u_i') &\overset{1}{=} f_0 \left( \sum_{j \in supp(u)} R_{i,j} u_j \right) \\
&\overset{2}{=} f_0 \left( \sum_{j \in supp(u) \cap \mathcal{N}(i)} R_{i,j} u_j \right) \\
&\overset{3}{=} f_0 \left( u_{j = supp(u) \cap \mathcal{N}(i)} \right) \\
&\overset{4}{\neq} 0
\end{aligned}
\tag{12}
$$

for the case of $i \in \mathcal{N}(supp(u))$. The derivation of (12) is detailed as follows: (i) The first equation results from the definition of $supp(u)$, which holds $u_i \neq 0$ for $i \in supp(u)$, and otherwise, $u_i \neq 0$. (ii) The second equation is deduced by Definition 1, that is $j \in \mathcal{N}(i)$, if $R_{i,j} \neq 0$. (iii) By the structure of $R \in \{0,1\}^{m \times n}$ with column degree $d$ and with $R_{*,\phi}^{\top} R_{*,\phi} = d I_{|\phi|}$, $\phi = supp(u) \cup supp(v)$, it is easy to see that the columns of $R_{*,\phi}$ are orthogonal to each other, and equivalently, $\mathcal{N}(j_1) \cap \mathcal{N}(j_2) = \emptyset$, $\forall j_1 \neq j_2$ and $j_1, j_2 \in \phi$ (or $\in supp(u) \subset \phi$); the orthogonality property suggests that there exists only one column index $j \in \mathcal{N}(i) \cap supp(u)$ (and satisfying $R_{i,j} = 1$), $\forall i \in \mathcal{N}(supp(u))$, and this yields the third equation. (iv) The fourth equation is easily derived by $u_j \neq 0$, $j \in supp(u)$.

Combing the results of (11) and (12), it follows that $supp(u') = \mathcal{N}(supp(u))$, which indicates that the support of the projected data $u'$ is the adjacent set of the support of the original data $u$. Similarly, the same result can also be derived for the other pair of data $v$, $v'$, that is $supp(v') = \mathcal{N}(supp(v))$. Then the support intersection $\psi'$ of the two projected data $u'$, $v'$ can be expressed as

$$
\begin{aligned}
\psi' &\overset{1}{=} supp(u') \cap supp(v') \\
&\overset{2}{=} \mathcal{N}(supp(u)) \cap \mathcal{N}(supp(v)) \\
&\overset{3}{=} \mathcal{N}(supp(u) \cap supp(v)) \\
&\overset{4}{=} \mathcal{N}(\psi)
\end{aligned}
\tag{13}
$$

which has the third equation derived by the orthogonality of $R_{*,\phi}$, implying $\mathcal{N}(j_1) \cap \mathcal{N}(j_2) = \emptyset$, $\forall j_1$, $j_2 \in \phi = supp(u) \cup supp(v)$. The result indicates that the support intersection $\psi'$ of projected data $u'$, $v'$ is identical to the adjacent set of the support intersection $\psi$ of original data $u$, $v$.

Given $\psi' = \mathcal{N}(\psi)$ in (13), we can further formulate (10) as

$$
\begin{aligned}
f_0(u')^{\top} f_0(v') &\overset{1}{=} \sum_{i \in \psi'} f_0(u_i') f_0(v_i') \\
&\overset{2}{=} \sum_{j \in \psi} \sum_{i \in \mathcal{N}(j)} f_0(u_i') f_0(v_i') \\
&\overset{3}{=} \sum_{j \in \psi} \sum_{i \in \mathcal{N}(j)} f_0(u_j) f_0(v_j) \\
&\overset{4}{=} d \cdot \sum_{j \in \psi} f_0(u_j) f_0(v_j) \\
&\overset{5}{=} d \cdot f_0(u)^{\top} f_0(v)
\end{aligned}
\tag{14}
$$

for which the derivation is detailed as follows. (i) The second equation is derived by the result of (13), that is $\psi' = \mathcal{N}(\psi) = \bigcup_{j \in \psi} \mathcal{N}(j)$, with $\mathcal{N}(j_1) \cap \mathcal{N}(j_2) = \emptyset$, $\forall j_1 \neq j_2$ and $j_1, j_2 \in \phi$. (ii) The third equation results from the uniqueness of $j \in \mathcal{N}(i) \cap supp(u)$, provided $i \in \mathcal{N}(j)$, $j \in \psi \subset supp(u)$; and the details can be found in the analysis of the third equation of (12). (iii) The fourth equation is derived by $\mathcal{N}(j) = d$. The proof is complete. □

### A.2 Proof of Theorem 2

*Proof.* The proof is similar to that of Theorem 1. First, we divide the element coordinates of the original data vectors $u$, $v$ into two groups in terms of their element quantization $f_{\tau_1}(u_i)$, $f_{\tau_2}(v_i)$ equal to zero or not, in order to define the support union $\phi = supp(f_{\tau_1}(u)) \cup supp(f_{\tau_2}(v))$ and the intersection $\psi = supp(f_{\tau_1}(u)) \cap supp(f_{\tau_2}(v))$. In the similar way, we further define the support union $\phi'$ and intersection $\psi'$ for the projected data $u'$, $v'$. Then we need to identify the relation between $\psi'$ and $\psi$. To achieve this, as in (11) and (12), we propose to determine the value of $f_{\tau_1}(u'_i)$ in terms of $i \in \mathcal{N}(supp(f_{\tau_1}(u)))$ or not. For the case of $i \notin \mathcal{N}(supp(f_{\tau_1}(u)))$, we have

$$f_{\tau_1}(u'_i) = f_0 \left( \sum_{j \in [n] \setminus supp(f_{\tau_1}(u))} R_{i,j} u_j \right) \tag{15}$$
$$= 0$$

since by the summation formula for geometric series, it can be deduced that $\tau_1 = \frac{|u^*_{k_1}| + |u^*_{k_1+1}|}{2}$ is greater than the absolute vale of the function input, under the condition of $|u^*_{i+1}|/|u^*_i| \le e^{-\beta}$ and $\beta \ge \ln(2 + \sqrt{3})$; and for the other case of $i \notin \mathcal{N}(supp(f_{\tau_1}(u)))$, we can derive

$$
\begin{aligned}
&f_{\tau_1}(u'_i) \\
&\stackrel{1}{=} f_{\tau_1} \left( \sum_{j \in supp(f_{\tau_1}(u))} R_{i,j} u_j + \sum_{j \in [n] \setminus supp(f_{\tau_1}(u_i))} R_{i,j} u_j \right) \\
&\stackrel{2}{=} f_{\tau_1} \left( u_{j=supp(f_{\tau_1}(u))) \cap \mathcal{N}(i)} + \sum_{j \in [n] \setminus supp(f_{\tau_1}(u_i))} R_{i,j} u_j \right) \\
&\stackrel{3}{=} f_{\tau_1} \left( u_{j=supp(f_{\tau_1}(u))) \cap \mathcal{N}(i)} \right) \\
&\stackrel{4}{\neq} 0
\end{aligned}
\tag{16}
$$

which has the third equation resulting from the relation of $\left| u_{j=supp(f_{\tau_1}(u))) \cap \mathcal{N}(i)} \right| > \left| \sum_{j \in [n] \setminus supp(f_{\tau_1}(u_i))} R_{i,j} u_j \right| + \tau_1$, while the relation can be derived using the same method as for (15). The above two results (15) and (16) are the major characteristics of the proof of Theorem 2, and the subsequent proof will proceed similarly as in Theorem 1, omitted here for brevity. □

### A.3 Proof of Theorem 3

*Proof.* The condition of $R^\top_{*,\phi} R_{*,\phi} = d I_{|\phi|}$ means that $R^\top_{*,j_1} R_{*,j_2} = 0$ for $\forall j_1, j_2 \in \phi$, $j_1 \ne j_2$. In other words, the nonzero entries of any two columns of $R_{*,\phi}$ have no coordinate overlapped. By the distribution of the nonzero entries, we can express the probability as

$$
\begin{aligned}
Pr\{R^\top_{*,\phi} R_{*,\phi} = d I_{|\phi|}\} &= \frac{C^d_m C^d_{m-d} \cdots C^d_{m-(|\phi|-1)d}}{(C^d_m)^{|\phi|}} \\
&= \frac{[(m-d)!]^{|\phi|}}{(m!)^{|\phi|-1}(m-|\phi|d)!}
\end{aligned}
$$

Given $m$ and $\phi$, define $g(d; m, \phi) = Pr\{R_{*,\phi}^\top R_{*,\phi} = dI_{|\phi|}\}$. Then it can be derived that

$$
\begin{aligned}
\frac{g(d; m, \phi)}{g(d+1; m, \phi)} &= \frac{\frac{[(m-d)!]^{|\phi|}}{(m!)^{|\phi|-1}(m-|\phi|d)!}}{\frac{[(m-(d+1))!]^{|\phi|}}{(m!)^{|\phi|-1}[m-|\phi|(d+1)]!}} \\
&= \frac{(m-d)^{|\phi|}}{\prod_{\ell=0}^{|\phi|-1}(m-|\phi|d-\ell)} \\
&> 1,
\end{aligned}
\tag{17}
$$

since $\frac{m-d}{m-|\phi|d-\ell} > 1$, given $|\phi| \geq 2$ and $0 \leq \ell \leq |\phi| - 1$. This indicates that $g(d; m, \phi)$ is a monotonically decreasing function with respect to $d$, with its maximum value attained at $d = 1$. Specifically, this maximum value is given by

$$
\begin{aligned}
g(d; m, \phi)|_{d=1} &= \frac{(m-1)!}{m^{|\phi|-1}(m-|\phi|)!} \\
&= \frac{\prod_{\ell=0}^{|\phi|-1}(m-\ell)}{m^{|\phi|}}.
\end{aligned}
\tag{18}
$$

The proof is complete. $\qquad\square$

### A.4    Other experimental results

### A.4.1    Distance preservation

In Figure 7, we calculate the distance variation rate by (8) for *binary* quantized projections. The results are consistent with our theoretical prediction: the distance variation rate of binary matrices tends to increase with the column degree $d$, particularly when the sparsity ratio $k/n$ of original data is sufficiently small.

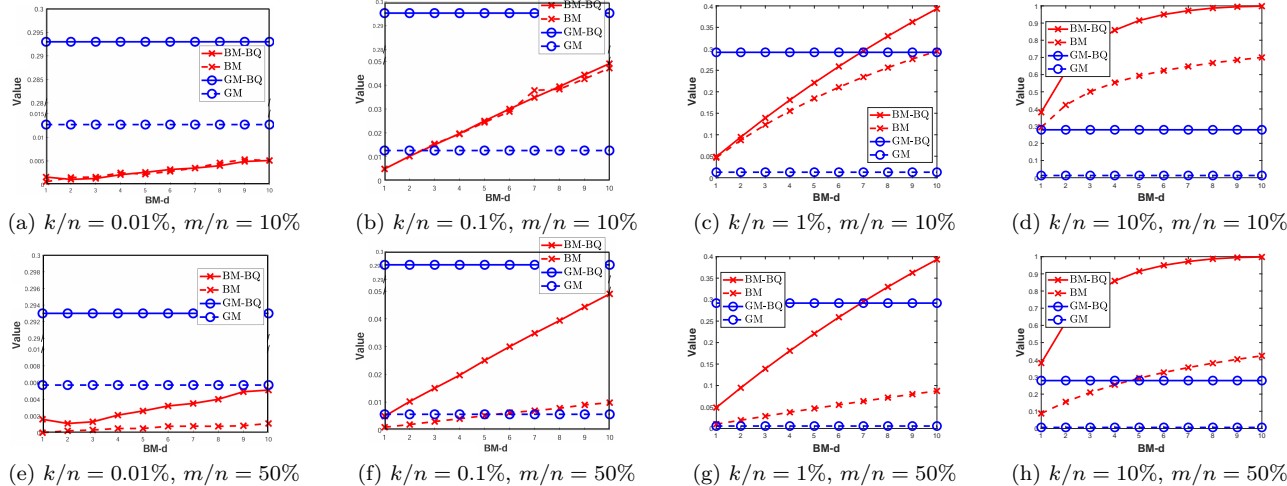

(a) $k/n = 0.01\%$, $m/n = 10\%$  (b) $k/n = 0.1\%$, $m/n = 10\%$  (c) $k/n = 1\%$, $m/n = 10\%$  (d) $k/n = 10\%$, $m/n = 10\%$

(e) $k/n = 0.01\%$, $m/n = 50\%$  (f) $k/n = 0.1\%$, $m/n = 50\%$  (g) $k/n = 1\%$, $m/n = 50\%$  (h) $k/n = 10\%$, $m/n = 50\%$

Figure 7: Distance variation rate for the binary-quantized (BQ) (and non-quantized) projections of the generated data, with four different feature sparsity ratios $k/n = 0.01\%$, $0.1\%$, $1\%$ and $10\%$, using two projection matrices: the Gaussian matrix (GM) and the binary matrix (BM) with varying column degree BM-d $\in [1, 10]$, under two projection ratios $m/n = 10\%$ and $50\%$. Note the smaller the distance variation rate, the better the distance preservation.

### A.4.2    Classification

In Figure 8, we conduct the **SVM** classification on the **ternary**-quantized projections of YaleB (DWT), and conduct the **KNN** classification on the **binary**-quantized projections of CIFAR10 (AlexNet) and Mini-ImageNet (VGG16), respectively, in Figures 9 and 10. The classification results in Figures 8-10 demonstrate a performance trend that aligns with our theoretical prediction: the extremely sparse binary matrix with column degree $d = 1$ can achieve superior or comparable performance to other denser matrices with larger $d$ values.

### A.4.3    Clustering

In Figures 11-13 and Figures 14-16, we respectively examine the k-means clustering performance on the **ternary** and **binary**-quantized projections of YaleB (DWT), CIFAR10 (AlexNet) and Mini-ImageNet (VGG16). The results align with our theoretical prediction: the extremely sparse binary matrix with column degree $d = 1$ can achieve superior or comparable performance to other denser matrices with larger $d$ values.

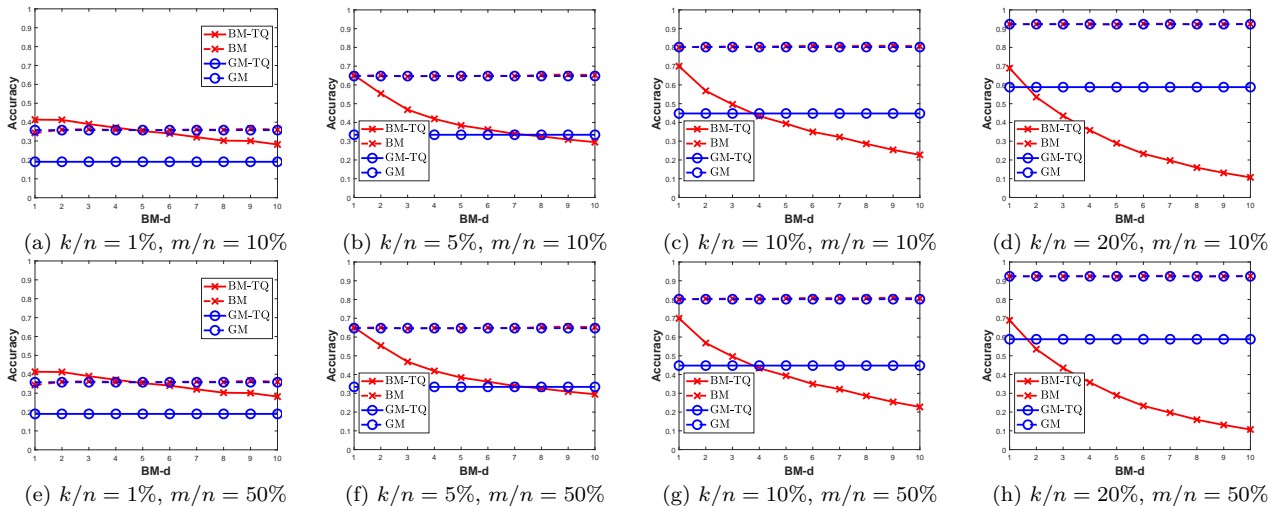

Figure 8: SVM classification accuracy for the ternary-quantized (TQ) (and non-quantized) projections of the exactly sparse features of YaleB (DWT), with four different feature sparsity ratios $k/n = 1\%$, $5\%$, $10\%$ and $20\%$, using two projection matrices: the Gaussian matrix (GM) and the binary matrix (BM) with varying column degree BM-d $\in [1, 10]$, under two projection ratios $m/n = 10\%$ and $50\%$.

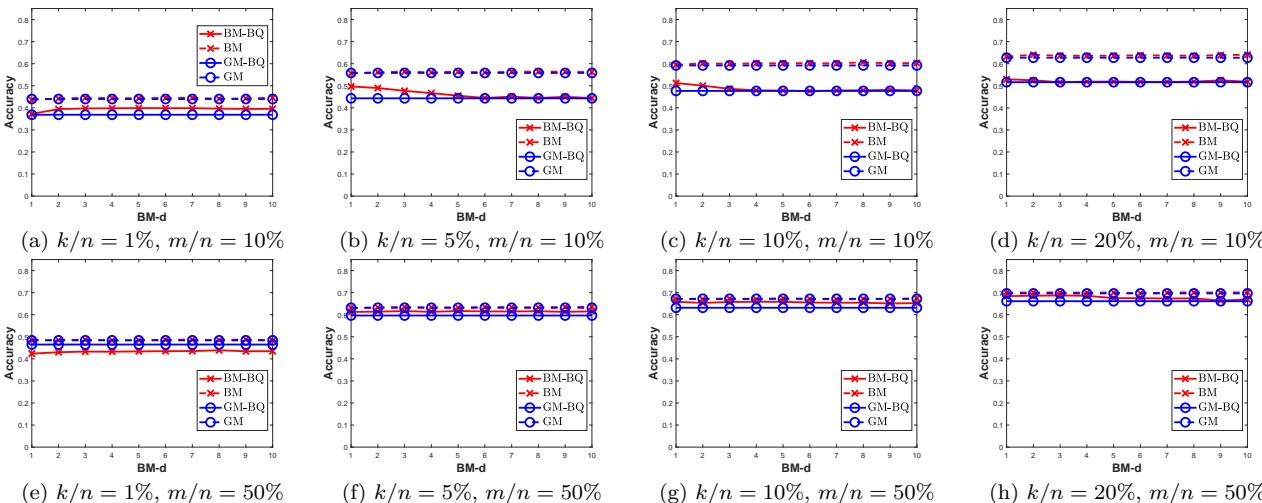

Figure 9: Classification accuracy for the binary-quantized (BQ) (and non-quantized) projections of the exactly sparse features of CIFAR10 (AlexNet), with four different feature sparsity ratios $k/n = 1\%$, $5\%$, $10\%$ and $20\%$, using two projection matrices: the Gaussian matrix (GM) and the binary matrix (BM) with varying column degree BM-d $\in [1, 10]$, under two projection ratios $m/n = 10\%$ and $50\%$.

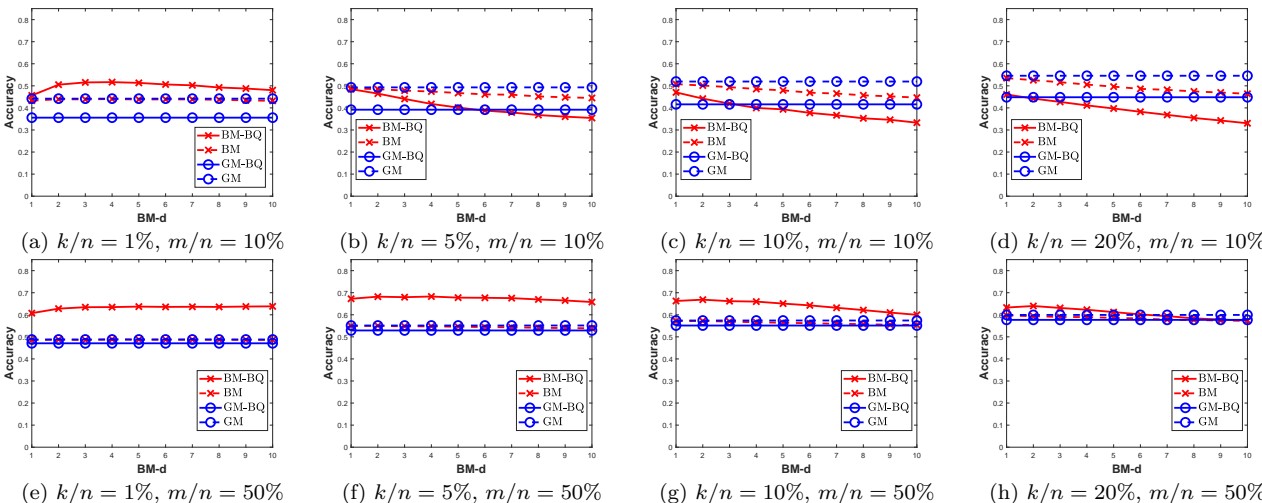

Figure 10: Classification accuracy for the binary-quantized (BQ) (and non-quantized) projections of the exactly sparse features of Mini-ImageNet (VGG16), with four different feature sparsity ratios $k/n = 1\%$, $5\%$, $10\%$ and $20\%$, using two projection matrices: the Gaussian matrix (GM) and the binary matrix (BM) with varying column degree BM-d $\in [1, 10]$, under two projection ratios $m/n = 10\%$ and $50\%$.

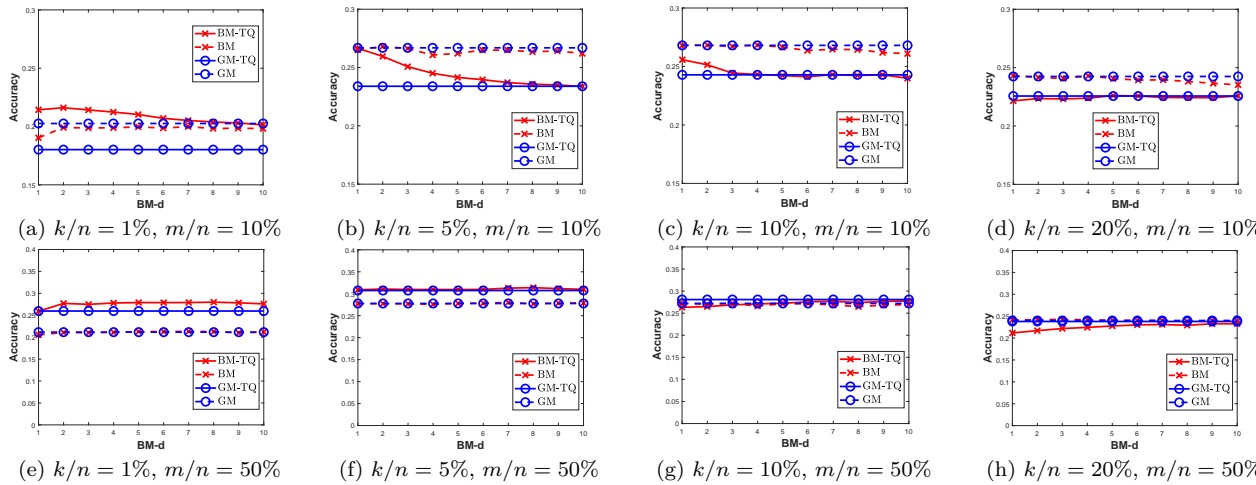

Figure 11: Clustering accuracy for the ternary-quantized (TQ) (and non-quantized) projections of the exactly sparse features of YaleB (DWT), with four different feature sparsity ratios $k/n = 1\%$, $5\%$, $10\%$ and $20\%$, using two projection matrices: the Gaussian matrix and the binary matrix with varying column degree BM-d $\in [1, 10]$, under two projection ratios $m/n = 10\%$ and $50\%$.

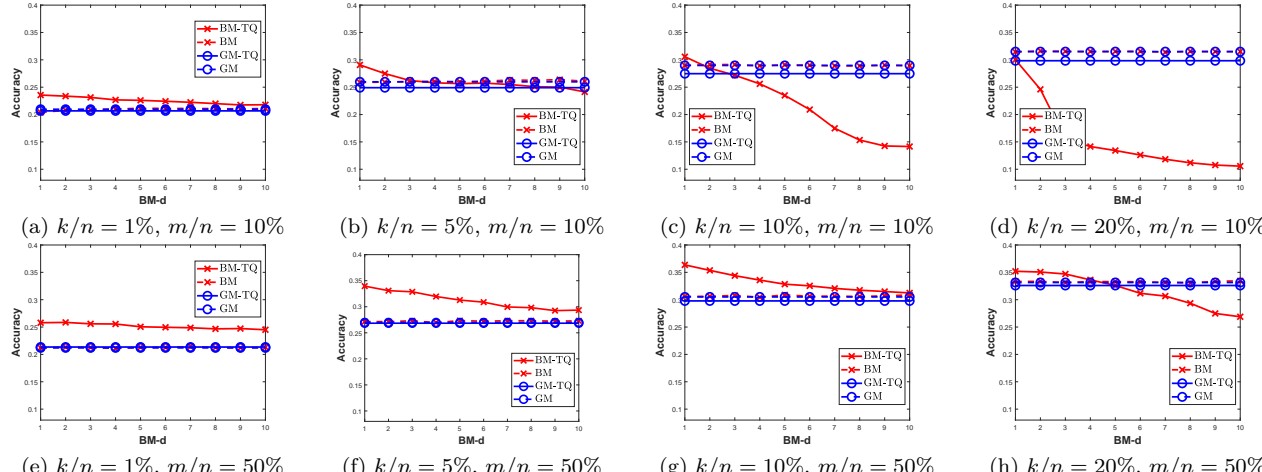

Figure 12: Clustering accuracy for the ternary-quantized (TQ) (and non-quantized) projections of the exactly sparse features of CIFAR10 (AlexNet), with four different feature sparsity ratios $k/n = 1\%$, 5%, 10% and 20%, using two projection matrices: the Gaussian matrix and the binary matrix with varying column degree BM-d $\in [1, 10]$, under two projection ratios $m/n = 10\%$ and 50%.

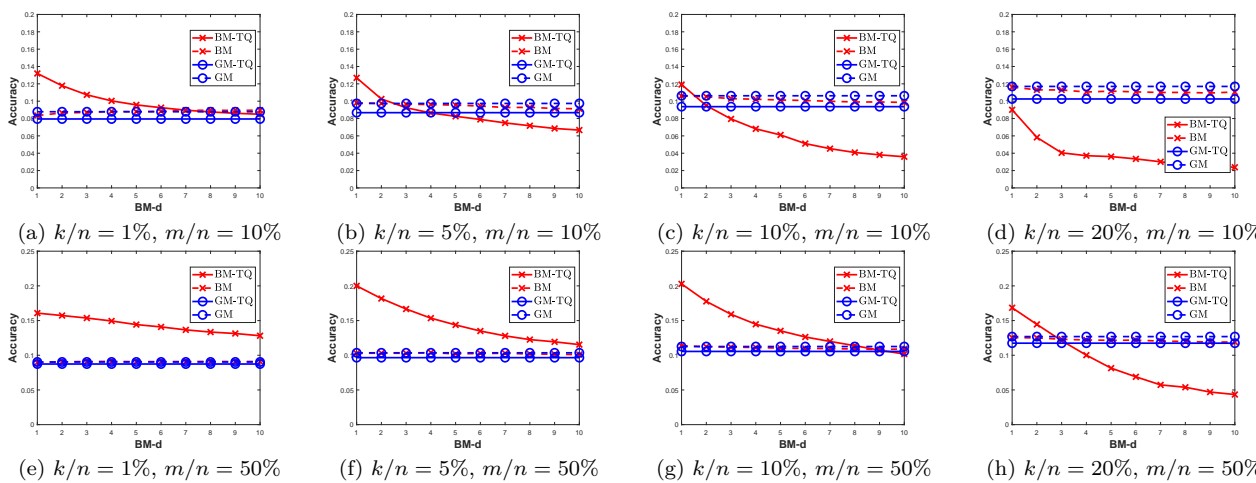

Figure 13: Clustering accuracy for the ternary-quantized (TQ) (and non-quantized) projections of the exactly sparse features of Mini-ImageNet (VGG16), with four different feature sparsity ratios $k/n = 1\%$, 5%, 10% and 20%, using two projection matrices: the Gaussian matrix and the binary matrix with varying column degree BM-d $\in [1, 10]$, under two projection ratios $m/n = 10\%$ and 50%.

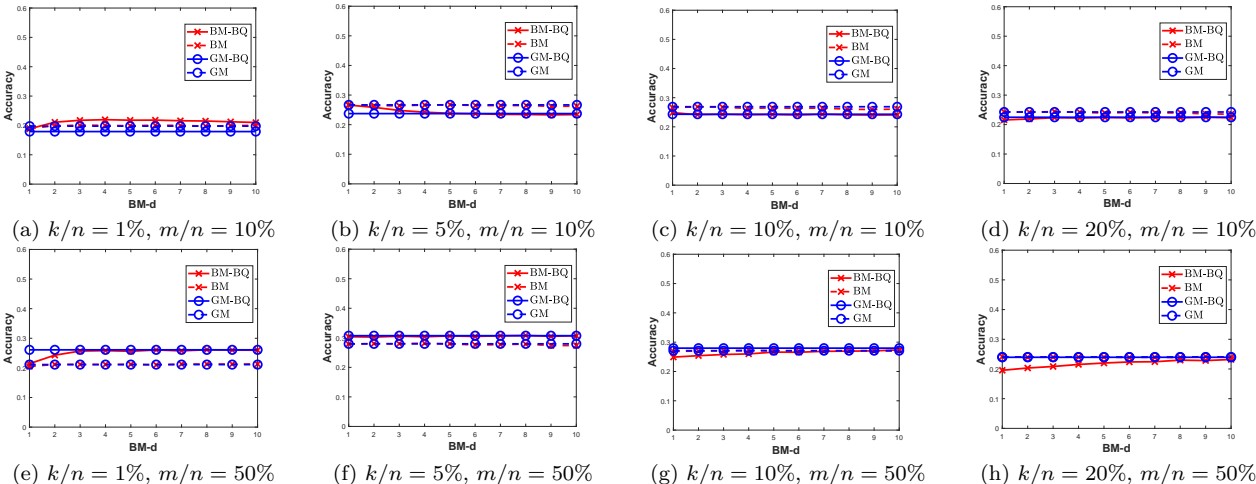

Figure 14: Clustering accuracy for the binary-quantized (BQ) (and non-quantized) projections of the exactly sparse features of YaleB (DWT), with four different feature sparsity ratios $k/n = 1\%$, $5\%$, $10\%$ and $20\%$, using two projection matrices: the Gaussian matrix and the binary matrix with varying column degree BM-d $\in [1, 10]$, under two projection ratios $m/n = 10\%$ and $50\%$.

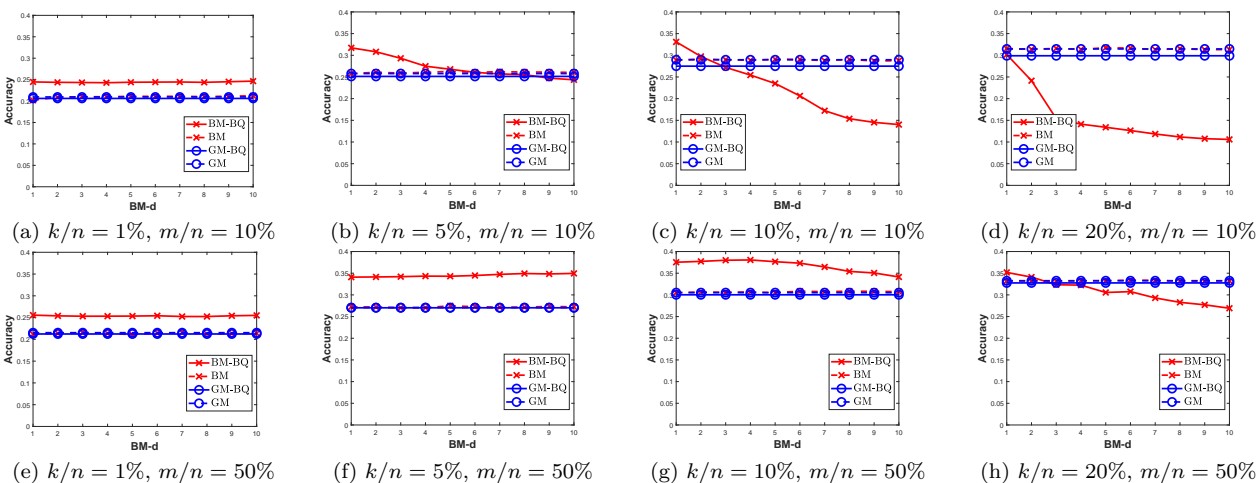

Figure 15: Clustering accuracy for the binary-quantized (BQ) (and non-quantized) projections of the exactly sparse features of CIFAR10 (AlexNet), with four different feature sparsity ratios $k/n = 1\%$, $5\%$, $10\%$ and $20\%$, using two projection matrices: the Gaussian matrix and the binary matrix with varying column degree BM-d $\in [1, 10]$, under two projection ratios $m/n = 10\%$ and $50\%$.

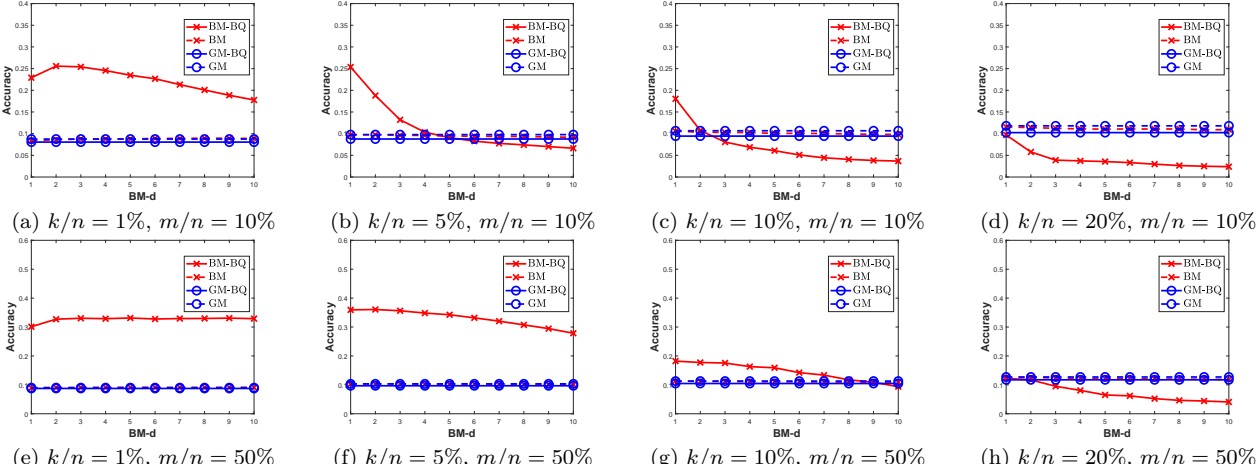

Figure 16: Clustering accuracy for the binary-quantized (BQ) (and non-quantized) projections of the exactly sparse features of Mini-ImageNet (VGG16), with four different feature sparsity ratios $k/n = 1\%$, $5\%$, $10\%$ and $20\%$, using two projection matrices: the Gaussian matrix and the binary matrix with varying column degree BM-d $\in [1, 10]$, under two projection ratios $m/n = 10\%$ and $50\%$.

