# OpenReview forum: "The Sparse Matrix-Based Random Projection:  A Study of Binary and Ternary Quantization"
_TMLR — Accepted by TMLR_

### Review · Reviewer_ov56 · 2024-11-06

**Summary Of Contributions:**

The paper studies the distance preservation properties of Random Projections on sparse quantized data. More specifically, it analyzes the case of projecting sparse feature data on which binary or ternary quantization was applied. On this later point, it differs from the current literature that mainly analyzed either random projections without quantization, or applying quantization after the projection.

First providing a theoretical analysis leading to estimating the optimal conditions and sparsity parameters (i.e. number of nonzero coefficients of the binary / ternary random projection matrix) to ensure distance preservation with high probability, the paper goes on to showing the results of some classification experiments run on benchmark datasets such as YaleB and CIFAR10.
These experiments demonstrate that sparse binary and ternary random projection matrices, when properly optimized, lead to comparable if not improved classification performance over denser matrices, all while offering reductions in computational and storage complexity.

**Audience:**

Yes

**Claims And Evidence:**

Yes

**Requested Changes:**

See weaknesses section. In short, I believe the paper should refocus the Experiment part on showcasing the distance preservation properties in practice - e.g. with analyzing distance matrices. The classification experiments can then be added as a use case but - as explained before - should not be the center of the empirical study. The paper could also study other Machine Learning tasks where distance based methods are popular to show the distance preservation properties in different settings.

Following the second weakness point, another requested change is adding more nuance in the analyses presented and discuss the limitations of quantization and random projections.

Final requested change: add code for reproducibility.

**Strengths And Weaknesses:**

Strengths:
- Paper is well structured and progression in the reasoning is clear and easy to follow
- Theoretical results on distance preservation are interesting and promising
- Rigorous analysis of the classification results, as the paper gives insights on each components of the method presented (e.g. Binary vs Gaussian matrices, quantized vs not quantized data)

Weaknesses:
- I believe that the paper's main weakness is its scope and methodology choice: the main results are on the distance preservation properties of Random Projections with quantized data, while the empirical experiments are done on classification algorithms. It feels that the reason experiments are being done on classification is because of distance based algorithms such as KNN - yet this is also true for a number of  other models and tasks. Therefore there is very little empirical analysis on distance preservation itself and the extensive work on classification seems peripheral.
- A second weakness of the paper is the absence of a tradeoff / limits discussion section: Although the paper highlights the benefits of sparse binary and ternary random projections for classification tasks, it does not delve deeply into the trade-offs involved, e.g. robustness or exposure to outliers. Therefore the paper lacks nuance.

---

> ### Author Response · Authors · 2024-12-17
> **Response to Reviewer ov56**
>
> Dear Reviewer ov56,
>
> Thank you for sparing your precious time to review our manuscript. In the revised manuscript, we have addressed all reviewer concerns, and have included additional numerical analyses in Section 4.4 (Figures 1 and 7), as well as clustering experiments in Section 5.3 (Figures 11-16) to validate our theoretical findings. The new or revised sections are indicated in blue. The code can be accessed at: \url{https://anonymous.4open.science/r/The-Sparse-Matrix-Based-Random-Projection-D13D}. Below we answer the comments one by one.
>
> **Comment 1:** I believe that the paper's main weakness is its scope and methodology choice: the main results are on the distance preservation properties of Random Projections with quantized data, while the empirical experiments are done on classification algorithms. It feels that the reason experiments are being done on classification is because of distance based algorithms such as KNN - yet this is also true for a number of other models and tasks. Therefore there is very little empirical analysis on distance preservation itself and the extensive work on classification seems peripheral.
>
> **Answer 1:** Thank you. Following the suggestion, in Section 4.4 we have directly investigated the impact of matrix sparsity on the distance preservation performance through numerical simulations. The results are provided in Figures 1 and 7, consist with our theoretical prediction. For details, please see the discussion in Section 4.4.
>
> **Comment 2:** A second weakness of the paper is the absence of a tradeoff/ limits discussion section: Although the paper highlights the benefits of sparse binary and ternary random projections for classification tasks, it does not delve deeply into the tradeoff involved, e.g. robustness or exposure to outliers. Therefore the paper lacks nuance.
>
> **Answer 2:** Thank you. In our study, we have theoretically derived that the binary matrix with column degree $d=1$ should outperform other denser matrices in distance preservation, thereby leading to better classification performance. This conclusion is  drawn on the assumption that the original data features intended for projection exhibit sufficiently sparse structures. Then for our model, noise mainly arises from the insufficient sparsity of the actual data features. To assess the impact of this noise, we have varied the feature sparsity ratio $k/n$ from 1\% to 100\% (as illustrated by the original features in Figure 6, Section 5.2.5) in the classification and clustering experiments. Notice the larger the $k/n$ value, the less sparse the data. Experimental results show that the binary matrix with $d=1$ can usually achieve superior or at least comparable performance to other denser matrices. This implies that our theoretical findings exhibit robust universal applicability, being relatively insensitive to the sparsity conditions imposed on the original data.
>
> **Comment 3:** The paper could also study other Machine Learning tasks where distance based methods are popular to show the distance preservation properties in different settings.
>
> **Answer 3:** Thank you. Following the suggestion, in Section 5.3 (Figures 11-16) we have further investigated the impact of matrix sparsity on the k-means clustering, a typical *unsupervised* leaning task. The results are consistent with our theoretical prediction: the binary matrix with $d=1$ tends to outperform other denser matrices. For more details, please see the discussion in Section 5.3.

---

### Review · Reviewer_MZJT · 2024-11-11

**Summary Of Contributions:**

The paper studies random projection with quantization using a binary matrix. The key motivation behind the setting is possible hardware accelerations due to the binary structure of the random projection matrix and the quantized embedding. Analyzing distance preservation before and after projection and quantization under random projection is standard. However, the paper analyzes preserving the distance between the quantized version of the input and the projection. The paper demonstrates that this new notion of distance preservation will be satisfied for projection matrices with sub-orthogonal matrices and then analyzes conditions under which a random binary matrix satisfies that property, concluding that sparser projections are better suited for distance preservation.

**Audience:**

Yes

**Broader Impact Concerns:**

There are no broader impact concerns.

**Claims And Evidence:**

Yes

**Requested Changes:**

- Based on the empirical results, the binary random projection matrix has accuracy comparable to the Gaussian matrix. It would be interesting to demonstrate the computational advantages of the binary projection in some real-world experiments. How much computational cost does the binary matrix buy in practice for classification?

- In the introduction, the contribution of Lu et al. 2023 should be discussed and contracted with the correct work because there seem to be some overlaps that can be explained.

- How does the sparse binary projection work for input vectors x that are not sparse and easily quantizable? It would be good to discuss potential regimes in which the new notion of distance preservation might not lead to preserving the original distance between data points.

- Since both the projection matrix and the data are very sparse, there is a risk of getting all zero outputs. Is this a valid concern?

**Strengths And Weaknesses:**

Strength

- Analyzing this new notion of distance preservation is novel

- The proof sketch is intuitive and well-explained

- The experiments support the claim and the theory


Weaknesses

- The proposed study would be relevant only for the settings where the quantization of the original input data f(x) is not too far from x. This limits the scope of the proposed projection.

---

> ### Author Response · Authors · 2024-12-17
> **Response to Reviewer MZJT**
>
> Dear Reviewer MZJT,
>
> Thank you for sparing your precious time to review our manuscript. In the revised manuscript, we have addressed all reviewer concerns, and have included additional numerical analyses in Section 4.4 (Figures 1 and 7), as well as clustering experiments in Section 5.3 (Figures 11-16) to validate our theoretical findings. The new or revised sections are indicated in blue. The code can be accessed at: \url{https://anonymous.4open.science/r/The-Sparse-Matrix-Based-Random-Projection-D13D}. Below we answer the comments one by one.
>
>
>
> **Comment 1:** The proposed study would be relevant only for the settings where the quantization of the original input data f(x) is not too far from x. This limits the scope of the proposed projection.
>
> **Answer 1:** Thank you. We would like to clarify that our theoretical analysis only requires the original input data vector $x$ to be adequately sparse, rather than being close to its ternary or binary quantization $f_{\tau}(x)$.
>
> In practical terms, meeting this condition is not particularly challenging, as the data features commonly studied, such as DWT, DCT and CNN, typically exhibit approximately sparse structures. When further transforming the approximately sparse structures to the exactly sparse structure required in our theoretical analysis, we usually can achieve improved classification performance, as evidenced in [r]. In addition, as discussed in the subsequent Answer 4, our theoretical findings also hold for the original  features, even when they do not precisely meet our sparsity conditions. This implies that our method has a broad range of applications.
>
> [r] Weizhi Lu, Mingrui Chen, Kai Guo, and Weiyu Li. Quantization: Is it possible to improve classification? In Data Compression Conference, pp. 318–327. IEEE, 2023.
>
>
>
>
> **Comment 2:** Based on the empirical results, the binary random projection matrix has accuracy comparable to the Gaussian matrix. It would be interesting to demonstrate the computational advantages of the binary projection in some real-world experiments. How much computational cost does the binary matrix buy in practice for classification?
>
> **Answer 2:** Thank you. Given a matrix size $m\times n$, it can be derived that binary matrices with column degree $d$ have computational complexity $O(dn)$, while Gaussian matrices have  complexity  $O(32mn)$ on a 32-bit precision platform.
>
> **Comment 3:** In the introduction, the contribution of Lu et al. 2023 should be discussed and contracted with the correct work because there seem to be some overlaps that can be explained.
>
> **Answer 3:** Thank you. Following the suggestion, we have discussed the above reference in Section 2 (related work), with the content highlighted in blue.
>
> **Comment 4:** How does the sparse binary projection work for input vectors x that are not sparse and easily quantizable? It would be good to discuss potential regimes in which the new notion of distance preservation might not lead to preserving the original distance between data points.
>
> **Answer 4:** Thank you. In our study, we have theoretically derived that the distance preservation capability of binary matrices tends to decline with the increasing of column degree $d$, if the original data features intended for projection exhibit sufficiently sparse structures. To further validate the result, in Section 4.4 we have directly investigated the impact of matrix sparsity on distance preservation through numerical simulations. The results in Figures 1 and 7 demonstrate that the performance trend described above is evident, when the original data have relatively small sparsity ratios $k/n$, such as $k/n<1\%$, corresponding to sufficiently sparse structures. This is consistent with our theoretical prediction.  As the sparsity ratio $k/n$ increases, it is found that binary matrices tend to exhibit similar distance preservation performance across different column degrees $d$.
>
> The above performance trend in distance preservation is corroborated in our classification and clustering experiments. Experimental results show that binary matrices with $d=1$ can usually achieve better or at least comparable performance to other denser matrices, even when $k/n$ reaches to 20\% (for exactly sparse features) or  to 100\% (for the original features shown in Figure 6, Section 5.2.5). This implies that our theoretical findings exhibit robust universal applicability, being relatively insensitive to the sparsity conditions imposed on the original data.
>
> **Comment 5:** Since both the projection matrix and the data are very sparse, there is a risk of getting all zero outputs. Is this a valid concern?
>
> **Answer 5:** Thank you. Theoretically, there is no risk of getting all zero outputs. This is because the sparse binary matrices we investigate have a  number $d$ of 1’s per column, ensuring that each element $x_i$ of the original data vector $\bf{x}$ will be sampled even when $d=1$.

---

> > ### Comment · Reviewer_MZJT · 2025-01-10
> > **Thanks**
> >
> > This addresses my concerns,

---

### Review · Reviewer_oRdZ · 2024-12-11

**Summary Of Contributions:**

Edited review:

This paper aims to study the preservation of inner products after a random projection and subsequent quantization of the input and output signal. The inner product is preserved because the random projection matrix is very specific: it has at most one 1 per row and exactly d 1s per column.

The second result is an adaptation to a setting where the signal decays and the threshold in the quantization is adapted to the decay of the signal. I didn't check the proof details but I believe it's correct.

The third result studies a random matrix's probability of having the specific property described above.

Note:
In a previous version of this review, I asked about a possible counter-example to the main theorem. The counter-example was incorrect due to a misunderstanding of the Theorem's hypothesis.

**Audience:**

Yes

**Broader Impact Concerns:**

None.

**Claims And Evidence:**

Yes

**Requested Changes:**

Release a github repository with the code to the experiments.

**Strengths And Weaknesses:**

Strengths:
The theoretical results seem correct.

Weaknesses:
In my opinion, the setting is too simplistic.

As per TMLR policy:
Are the claims made in the submission supported by accurate, convincing and clear evidence?
Yes.
Would at least some individuals in TMLR's audience be interested in knowing the findings of this paper?
Maybe, I don't know.

---

> ### Author Response · Authors · 2024-12-12
>
> Dear Reviewer oRdZ，
>
> Thank you for sparing your precious time to review our manuscript. We found a mistake in the above example, which incorrectly examined a $\\{0,\pm 1\\}$-**ternary** matrix $R$  rather than the $\\{0, 1\\}$-**binary** matrix discussed in our theorems.
>
> To rectify this, we can modify the ternary matrix $R$ to be a binary one by changing its *third* and *fourth* rows to $\\{1,0,\*,\*,\*\\}$ and $\\{0,1,\*,\*,\*\\}$, respectively, under the theoretical constraint $R_{\*,\phi}^\top R_{\*,\phi}=d I_{|\phi|}$. This will result in $f_0(u')^\top f_0(v') = 0=f_0(u)^\top f_0(v) $, aligning with our theoretical analysis.
>
> Additionally, we have addressed all concerns raised by the other two reviewers, **MZJT** and **ov56**, and will submit our responses and the revised manuscript shortly. The revised version includes numerical simulations to further validate our theoretical findings.
>
> Thank you once again.
>
> Sincerely,
>
> The authors

---

> ### Author Response · Authors · 2024-12-17
> **Response to Reviewer oRdZ**
>
> Dear Reviewer oRdZ,
>
> Thank you for sparing your precious time to review our manuscript. In the revised manuscript, we have addressed all reviewer concerns, and have included additional numerical analyses in Section 4.4 (Figures 1 and 7), as well as clustering experiments in Section 5.3 (Figures 11-16) to validate our theoretical findings. The new or revised sections are indicated in blue. The code can be accessed at: \url{https://anonymous.4open.science/r/The-Sparse-Matrix-Based-Random-Projection-D13D}. Below we answer the comments one by one.
>
>
>
> **Comment 1:**  In my opinion, the setting is too simplistic.
>
> **Answer 1:** Thank you. In our understanding, the “too simplistic” may refer to our theoretical requirement that the original input data need to be sufficiently sparse, either with exactly sparse structures (in Theorem 1) or with element magnitude decaying with high decay speed (in Theorem 2). We would like to clarify that these conditions are sound and not challenging to meet in practice, as the data features commonly studied, such as DWT, DCT and CNN, typically exhibit approximately sparse structures. When further transforming the approximately sparse structures to the exactly sparse structure required in Theorem 1, we usually can achieve comparable or improved classification performance, as evidenced in [r]. In addition, as discussed in Section 5.2.5, our theoretical findings also hold for the original features without exactly sparse structures, even when they do not precisely meet the requirement of Theorem 2. This implies that our theoretical findings exhibit robust universal applicability, being relatively insensitive to the sparsity conditions imposed on the original data.
>
> [r] Weizhi Lu, Mingrui Chen, Kai Guo, and Weiyu Li. Quantization: Is it possible to improve classification? In Data Compression Conference, pp. 318–327. IEEE, 2023.
>
>
> **Comment 2:**  As per TMLR policy: Are the claims made in the submission supported by accurate, convincing and clear evidence? Yes. Would at least some individuals in TMLR's audience be interested in knowing the findings of this paper? Maybe, I don't know.
>
> **Answer 2:** Thank you. For binary matrix-based random projection, we have investigated the impact of matrix sparsity on the ternary/binary quantized projections. This issue holds fundamental significance in machine learning, as many models incorporate random projections or similar operations for dimensionality reduction or evaluating linear feature similarity. Examples include hash coding in large-scale retrieval systems or individual layers in deep neural networks. Our research reveals that these models’ parameters may be quantized to extremely spare binary/ternary values without sacrificing performance, thereby prompting further exploration into potential quantization methods.

---

### Decision · Action_Editor_inPy · 2025-01-23

**Recommendation:** Accept as is

**Comment:**

The paper is on topic for TMLR, given that it applies a variation of a key tool (sparse random projections), and evaluates for some distance-based ML problems.  The paper includes technical analysis, and empirical evidence that the proposed approach is successful.  While there were some (relatively minor) concerns from the reviewers, the authors have addressed them in the updated paper.

**Audience:**

Sparse projection and quantization, with demonstration in distance-based ML tasks is within the scope of TMLR.

**Claims And Evidence:**

The reviewers agree that the paper's technical results are correct, and the empirical evidence is sufficient.